# The proline-rich antimicrobial peptide Api137 disrupts large ribosomal subunit assembly and induces misfolding

Simon Malte Lauer [1,2,8], Jakob Gasse [3,4,8], Andor Krizsan[3,4], Maren Reepmeyer [3,4], Thiemo Sprink[5,6], Rainer Nikolay [1,7] ✉, Christian M. T. Spahn [1] ✉ & Ralf Hoffmann [3,4] ✉

The proline-rich antimicrobial designer peptide Api137 inhibits protein expression in bacteria by binding simultaneously to the ribosomal polypeptide exit tunnel and the release factor (RF), depleting the cellular RF pool and leading to ribosomal arrest at stop codons. This study investigates the additional effect of Api137 on the assembly of ribosomes using an *Escherichia coli* reporter strain expressing one ribosomal protein per 30S and 50S subunit tagged with mCherry and EGFP, respectively. Separation of cellular extracts derived from cells exposed to Api137 in a sucrose gradient reveals elevated levels of partially assembled and not fully matured precursors of the 50S subunit (pre-50S). High-resolution structures obtained by cryogenic electron microscopy demonstrate that a large proportion of pre-50S states are missing up to five proteins (uL22, bL32, uL29, bL23, and uL16) and have misfolded helices in 23S rRNA domain IV. These data suggest a second mechanism for Api137, wherein it disrupts 50S subunit assembly by inducing the formation of misfolded precursor particles potentially incapable of evolving into active ribosomes, suggesting a bactericidal mechanism.

The near-universal emergence of antibiotic resistance is regarded as one of the greatest threats to human health that, if not addressed, could cause more deaths from bacterial infections than cancer by 2050[1]. The current annual cost of infections caused by antibiotic-resistant bacteria in terms of impact on human healthcare has been estimated to be more than US $4 billion[2]. There is a consensus that new selective antimicrobial agents that overcome resistance mechanisms are urgently needed[3]. Proline-rich antimicrobial peptides (PrAMPs) are potential alternatives to small-molecule antibiotics due to their potency against many Gram-negative bacteria[4]. Insect-derived PrAMPs[4,5], including apidaecins expressed in

*Apis mellifera* (honeybees)[6], drosocin identified in *Drosophila melanogaster*[7], and oncocin derived from *Oncopeltus* antibacterial peptide 4 isolated from *Oncopeltus fasciatus* (milkweed bug)[8,9], are produced in response against bacterial infections. Despite structural similarities among PrAMPs expressed in different insects, they bind to the heat shock protein DnaK, which was identified as the first intracellular target[10,11], and the bacterial (70S) ribosome, which was later identified as the lethal target[12]. Although all insect-derived PrAMPs appear to bind to the bacterial ribosome, they occupy different regions in the ribosome and inhibit protein translation by different mechanisms[13–18].

[1]Institut für Medizinische Physik und Biophysik, Charité - Universitätsmedizin Berlin, corporate member of Freie Universität Berlin and Humboldt Universität zu Berlin, Berlin, Germany. [2]Humboldt-Universität zu Berlin, Institut für Biologie, 10099, Berlin, Germany. [3]Institute of Bioanalytical Chemistry, Faculty of Chemistry and Mineralogy, Universität Leipzig, Leipzig, Germany. [4]Center for Biotechnology and Biomedicine, Universität Leipzig, Leipzig, Germany. [5]Core Facility for Cryo-Electron Microscopy, Charité - Universitätsmedizin Berlin, Berlin, Germany. [6]Cryo-EM Facility, Max Delbrück Center for Molecular Medicine in the Helmholtz Association, Berlin, Germany. [7]Department of Genome Regulation, Max Planck Institute for Molecular Genetics, Berlin, Germany. [8]These authors contributed equally: Simon Malte Lauer, Jakob Gasse. ✉e-mail: nikolay@molgen.mpg.de; christian.spahn@charite.de; bioanaly@rz.uni-leipzig.de

The ribosome is a finely tuned macromolecular machine whose primary function, the mRNA-based production of polypeptides, is conserved across all domains of life[19,20]. While the overall function of the ribosome is the same in bacteria, archaea, eukaryotes, and eukaryotic organelles, the sequences of ribosomal components, such as ribosomal RNAs (rRNAs) and ribosomal proteins (RPs), differ significantly among domains of life[21]. Comparison of ribosome structures from different domains of life revealed that some RPs are universally conserved, and others are specific to bacteria or archaea and eukaryotes[20,21]. These compositional subtleties may account for deviations in the assembly process of ribosomes from different domains of life[22] and render ribosome assembly an attractive target for antimicrobial agents.

PrAMPs are evolutionarily optimized against the microbes attacking the host, but these native peptides are not adapted to therapeutic treatments. We have improved the pharmacological properties of apidaecin 1b, resulting in lead compounds Api88 and Api137[23,24]. Similarly, oncocin was optimized, leading to Onc72 (formerly O23) and Onc112 (formerly O24)[9,25]. All four designer peptides are highly efficient against Gram-negative bacteria in murine infection models[23,26–30]. They enter bacteria primarily through the SbmA transporter[31] and inhibit protein translation by binding to the ribosomal polypeptide exit tunnel (PET)[12,13,18]. Using high-resolution cryogenic electron microscopy (cryo-EM), various groups showed that Onc112 binds deep in the PET, blocking and destabilizing the initiation complex[15,32], while Florin et al. showed that Api137 binds to the PET near the peptidyl transferase center and inhibits protein translation by trapping release factors, thereby arresting ribosomes at stop codons[14]. Recently, we identified an additional binding site for Api137 within the 50S subunit, next to the exit pore of the PET, and a further binding pocket deep within domain III of the 23S rRNA, which is only occupied by Api88[33].

Surprisingly, a second mode of action for Api137 was reported in 2015. The appearance of an additional peak in sucrose gradient profiles adjacent to the large ribosomal subunit (50S)[13] raised the possibility that Api137 interferes with 50S assembly; however, this has not been further investigated. Recently, we combined the in vitro reconstitution assay of the 50S subunit with a cryo-EM approach and revealed the stepwise maturation of pre-50S precursors at near-atomic resolution, demonstrating the initial binding of the early assembly proteins uL24, uL29, and uL22[34,35]. The parallel routes of assembly and key incorporation events of RPs visualize the principle of cooperative binding of RPs during 50S assembly. Precursors of 50S samples isolated from bacteria after depletion of RPs[36] and depletion of ribosome assembly factors (AFs), which assist ribosome assembly without being part of mature ribosomal subunits, provided insights into the assembly process using biochemical approaches, mass spectrometry, and cryo-EM[37–42]. The only known small molecule reported with a direct inhibitory effect on ribosome assembly affecting 50S and the small ribosomal subunit (30S) is lamotrigine, curiously a drug used to treat epilepsy and schizophrenia[43]. Antibiotics that specifically and directly interfere with the assembly of either the 30S or 50S subunits have not been reported so far.

Here, we show that E. coli MC4100 cells without and with fluorescently labeled ribosomes[44,45] incubated with Api137 are growth-inhibited and accumulate precursors of the 50S subunit. Conventional sucrose gradients reveal a slower migrating additional peak, which contains precursors of the 50S subunit as identified by additional fluorescence analysis. Subsequent cryo-EM combined with single particle analysis of the isolated precursors reveals several states of incompletely assembled 50S subunits. While certain states exhibit similarities to previously characterized intermediates[34–36,41,46], surprisingly, nearly half of these precursors lack the early assembly protein uL22. This deficiency induces precursors not observed before, including a 180° misaligned helix (H61) within the 50S subunit, as a result of Api137-mediated assembly inhibition. The finding that

antimicrobial molecules, such as apidaecin peptides, can induce misfolding of ribosomal components during assembly offers an attractive avenue for future research and potential drug development.

## Results

### Api137 induces disruption in 50S ribosomal subunit assembly in *E. coli*

In a previous study[13], we showed that treatment of *E. coli* BL21(DE3)RIL with Api137 resulted in an extra peak between the 30S and 50S subunits in the $A_{254}$ ribosome profiles, suggesting the accumulation of a precursor of the 50S subunit. In contrast, treatment with Onc112 did not show any alterations and was therefore used as a control peptide in this study. With an adjusted workflow (Fig. 1a) we were able to confirm the effects with *E. coli* MC4100 (Fig. 1b) and to validate this observation we generated an MC4100-based *E. coli* reporter strain (RN31) with fluorescently tagged ribosomal subunits (Fig. 1c). Using gene knock-in strategies[44,45], RN31 was engineered to produce the ribosomal proteins bS20 and bL19, which are tagged at their surface-exposed C-terminus with mCherry and EGFP, respectively. A comprehensive characterization of RN31 confirmed the production of ribosomal protein fusions (with mCherry and EGFP) by mass spectrometry (https://panoramaweb.org/Api137_immature_ribo.url) and by the expected apparent molecular weight observed by SDS-PAGE (Fig. 1d). Importantly, the growth rates of RN31 in the absence and presence of the AMPs Onc112 and Api137, as well as the MICs for Onc112 (2 μg/mL) and Api137 (4 μg/mL), were comparable to those of the parental strain MC4100 (Fig. 1e). The $A_{254}$ ribosome profiles of *E. coli* strains MC4100 and RN31 cultured in the absence of a PrAMP were highly similar (Fig. 1b (black curve) and 1f). While Onc112 treatment (8 μg/mL) had little effect on both absorbance and fluorescence profiles (Fig. 1b (blue curve) and 1g), the presence of Api137 at the same concentration resulted in a significant increase in the 30S peak and a left-hand shoulder of the 50S, identified as precursors of the 50S subunit by green fluorescence readout (Fig. 1b (orange curve) and 1h). We note that our strategy detects only those precursors that have already incorporated the tagged proteins. Based on previous studies[35,47], both bL19 and bS20 assemble early during the biogenesis of the large and small subunit, respectively. Hence, this setup allowed monitoring and quantification of potential precursors of both the 50S and 30S ribosomal subunits. For precursors of the 30S subunit, however, we found no evidence.

Quantitative changes in ribosome composition upon Api137 treatment (Fig. 2) were assessed by integrating the absorbance and mCherry and EGFP fluorescence obtained for the combined fractions of the 30S subunit (fractions 30–43), 50S subunit (54–63), 70S ribosome (64–77), and the pre-50S region (44-53) between the 30S and 50S subunits, which presumably represents partially assembled 50S subunits. Mature 70S ribosomes and 50S subunits were essentially unaffected by Api137 treatment. The pre-50S region showed significantly higher absorbance and EGFP fluorescence levels, suggesting an accumulation of immature pre-50S subunits. The mCherry fluorescence was also slightly elevated in this region, probably due to a fronting effect resulting from the elevated 30S subunit peak. These higher absorbance and fluorescence levels indicated that the 30S subunits could not participate in 70S formation, probably due to the lack of fully assembled 50S subunits. Interestingly, there was also a significant increase in EGFP fluorescence, which may indicate increased levels of even smaller pre-50S subunits present in the region normally occupied by 30S subunits.

In addition, $A_{254}$ ribosome profiles with subsequent fluorometric analysis of the sucrose fractions demonstrated that treatment of RN31 with various ribosome-directed antibiotics resulted in detectable and expected differences in its ribosome assembly landscape (Supplementary Fig. 1). When RN31 was incubated with either Api88, the C-terminally amidated derivative of Api137, at the same concentration, or with the antibiotics chloramphenicol or erythromycin, only Api88

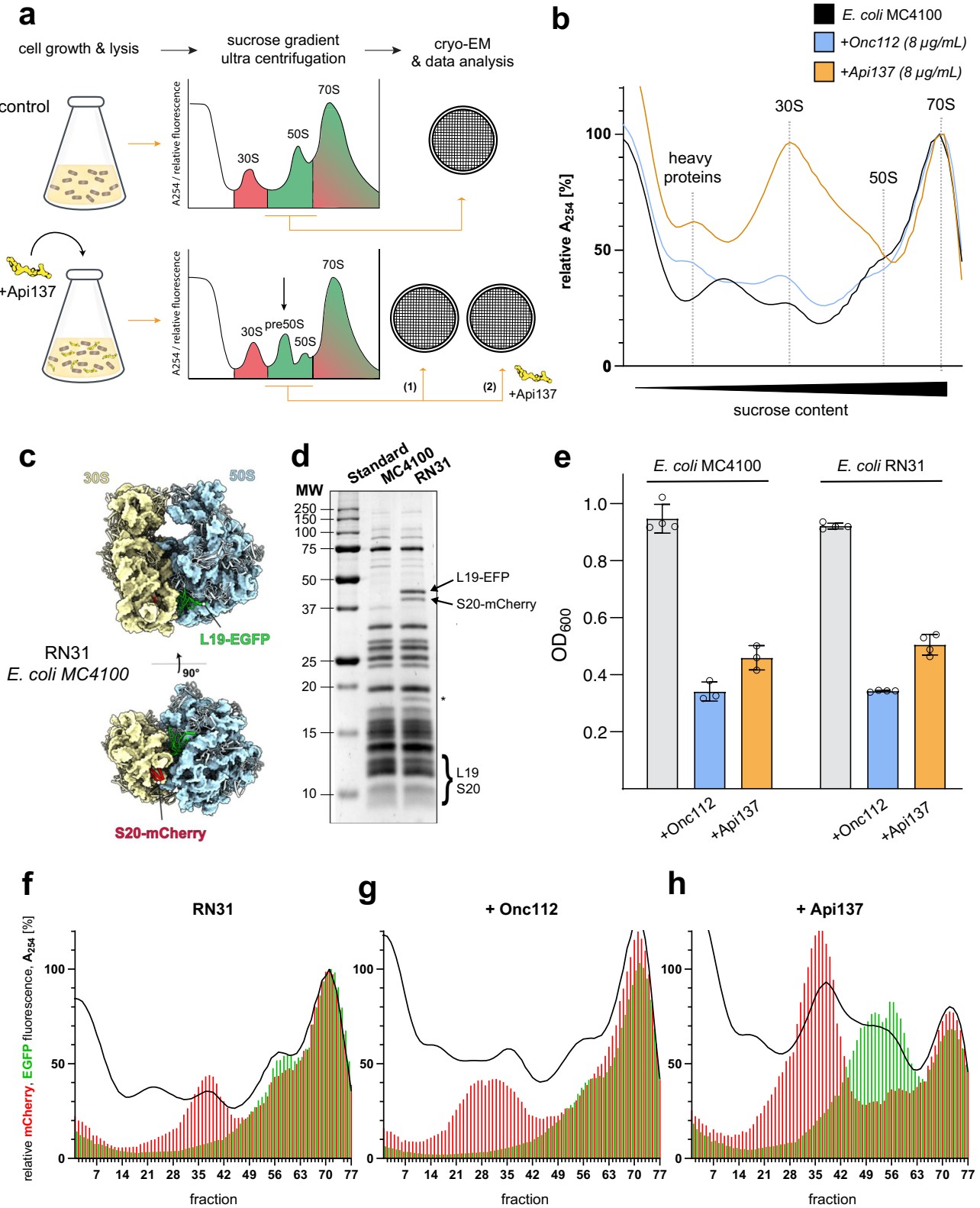

showed a slight accumulation of pre-50S precursors, whereas such precursors were not observed for the antibiotics chloramphenicol and erythromycin (Supplementary Fig. 1). Taken together, fluorometric analysis of ribosome profile fractions derived from RN31 after Api137 treatment revealed the substantial accumulation of incompletely assembled pre-50S ribosomal subunits. In contrast, treatment with the PrAMPs Api88 and Onc112 or ribosome-directed antibiotics did not result in the significant accumulation of immature pre-50S precursors.

## Api137 treatment induces structurally distinct pre-50S precursors

Sucrose gradient fractions 44 to 63, representing the pre-50S and 50S material obtained from *E. coli* RN31 cultures grown in the absence or presence of Api137, underwent ultrafiltration for buffer exchange and concentration. The ribosomal particles were then analyzed by cryo-EM, yielding the control dataset and the Api137 dataset, respectively.

**Fig. 1 | Ribosome composition and growth of *E. coli* strains MC4100 and RN31.**
**a** The general workflow for the study. **b** Ribosome absorbance profiles derived from *E. coli* MC4100 grown in culture medium without a PrAMP (black, *n* = 3), with Onc112 (blue, *n* = 3), or Api137 (yellow, *n* = 3) at a concentration of 8 μg/mL. All absorbances were normalized to the maximum peak intensity of the 70S fraction in the control. **c** Schematic of ribosomal subunits of RN31 tagged with fluorescent proteins. **d** SDS-PAGE (*T* = 16%) of 70S ribosomes isolated from both *E. coli* strains by sucrose cushion. The proteins bL19 (13.1 kDa) and bS20 (9.7 kDa) were detected at apparent molecular weights of 10 to 15 kDa only in *E. coli* MC4100, whereas bL19-EGFP and bS20-mCherry appeared as two separate bands at an apparent molecular weight of ~40 kDa only in RN31, which was confirmed by mass spectrometry (https://panoramaweb.org/Api137_

immature_ribo.url). The asterisk indicates an mCherry cleavage product[77]. The gel is representative of three independent repetitions. **e** $OD_{600}$ values were obtained for *E. coli* MC4100 (*n* = 3) and RN31 (*n* = 4) cell cultures grown in the exponential growth phase (*n* ≥ 3) for 90 min in the absence (control) or presence of Onc112 or Api137 (8 μg/mL) at 37 °C. PrAMPs were added to the cultures at an $OD_{600}$ of 0.2. **f–h** Absorbance ($A_{254}$, black curve) and EGFP and mCherry fluorescence values (green and red bars, respectively) obtained for *E. coli* RN31 grown in culture medium (**f**) without PrAMP (*n* = 10) or incubated with 8 μg/mL (**g**) Onc112 (*n* = 6) or (**h**) Api137 (*n* = 10) along a sucrose gradient. All absorbance and fluorescence values were normalized to the mean value of the corresponding maximum peak intensity of the 70S fraction in the control. Source data are provided as a Source Data file.

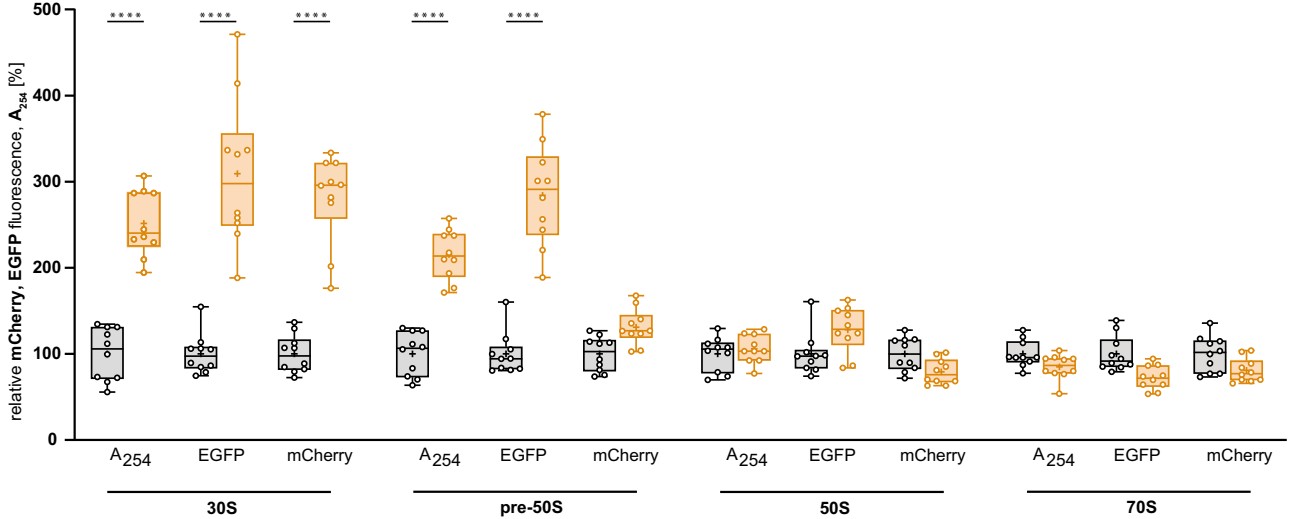

**Fig. 2 | Statistical analysis of absorbance ($A_{254}$) and fluorescence-based ribosome profiles of *E. coli* RN31 in the absence or presence of Api137.** Box plot showing the sum of the values recorded for the 30S (30–43), pre-50S (44–53), 50S (54–63), and 70S regions (fractions 64–77) of *E. coli* RN31 incubated with Api137 (8 μg/mL, yellow) relative to the mean of the untreated *E. coli* RN31 (gray) in the corresponding region. For both, treated and control samples, ten (*n* = 10) individual biological replicates were used for the statistics. Whiskers are shown min to max, while the hinges of the box show 25%–75% of the distribution. The mean is represented by a "+" and the median as a horizontal line. Statistical significance was determined by ordinary one-way ANOVA using Šídák's multiple comparisons test (****$p < 0.0001$). Source data are provided as a Source Data file.

At this point, a word of caution is warranted. The absence of density in a cryo-EM map does not necessarily indicate the absence of the corresponding component, because it could be present in a potentially flexible or dynamic state. When we describe the binding or docking of a protein or an rRNA element to a precursor particle, we acknowledge the possibility that it may have been previously associated with the particle. In addition, cryo-EM density for EGFP is not observed because the EGFP molecule is attached to the surface-exposed C-terminus of bL19 via a flexible linker. This results in a high degree of mobility, which precludes its detection using cryo-EM.

After an initial alignment and several rounds of variability sorting, seven ribosomal states common to both datasets were identified, i.e., 70S, 50S, and 30S ribosomal states and four pre-50S precursor states (Fig. 3, Supplementary Fig. 2–6). These precursors represent late stages of 50S assembly, with most of the 23S rRNA regions stably formed, while the subunit´s so-called functional core (FC) appeared unstructured. The FC comprises the peptidyl transferase center (PTC), located within the central bulge of domain V, and adjacent rRNA elements[48]. Two states had the central protuberance (CP) attached, whereas it was absent in the other two states.

Notably, Api137 treatment resulted in the accumulation of 11 additional pre-50S states (states 1-9 and 14-15) (Fig. 3a). Moreover, we observed significant changes in the relative particle numbers of ribosomal subunits and precursor states between the control and the Api137 dataset (Fig. 3b). The Api137 dataset contained less 70S particles (control: 26.1%, Api137: 15.4%) and 50S subunits (control: 40.9%, Api137: 18.0%), while the amount of 30S subunits (control: 23.7%,

Api137: 20.8%) remained relatively constant (Fig. 3c), which is in good agreement with fractions 44-63 of the ribosome profiles (Fig. 1f and h). We note that the number of particle images in cryo-EM classes is only a rough estimate of the relative populations in the sample. Some structural states may have a preferred orientation or excessive structural flexibility, preventing them from separating into 3D classes and being accurately counted. Nevertheless, there was a significant increase in the abundance of pre-50S precursors in the Api137 dataset, which represented 45.9% of the assigned particles, while the control contained only 9.2% of pre-50S intermediates. The identified particle states differed concerning their structural completeness, both regarding 50S ribosomal proteins (L-proteins) and 23S rRNA structural regions. Based on these structural distinctions, we categorized the 11 Api137 specific states into four groups representing different stages of maturation (Fig. 3), five of which lacked cryo-EM density for the CP, while six had docked the 5S rRNA, with the CP accordingly formed. The nomenclature we use for naming the individual states is based on the nomenclature recently introduced by Qin et al.[35]. In brief, each state is named according to the 23S rRNA domain for which it exhibits density. After the core of the subunit has formed, the name contains a *C*. In addition, L-proteins and structural features are denoted in the names. In cases where cryo-EM density for L-proteins is lacking, this absence is indicated by brackets and a minus symbol in the exponent (L-protein)⁻. One state (*d126_(L29)⁻/(L22)⁻*) represents an early assembly precursor prior to the formation of the subunit's core. The remaining states were either equivalent to or more mature than the core state, mostly lacking cryo-EM density for domain IV and

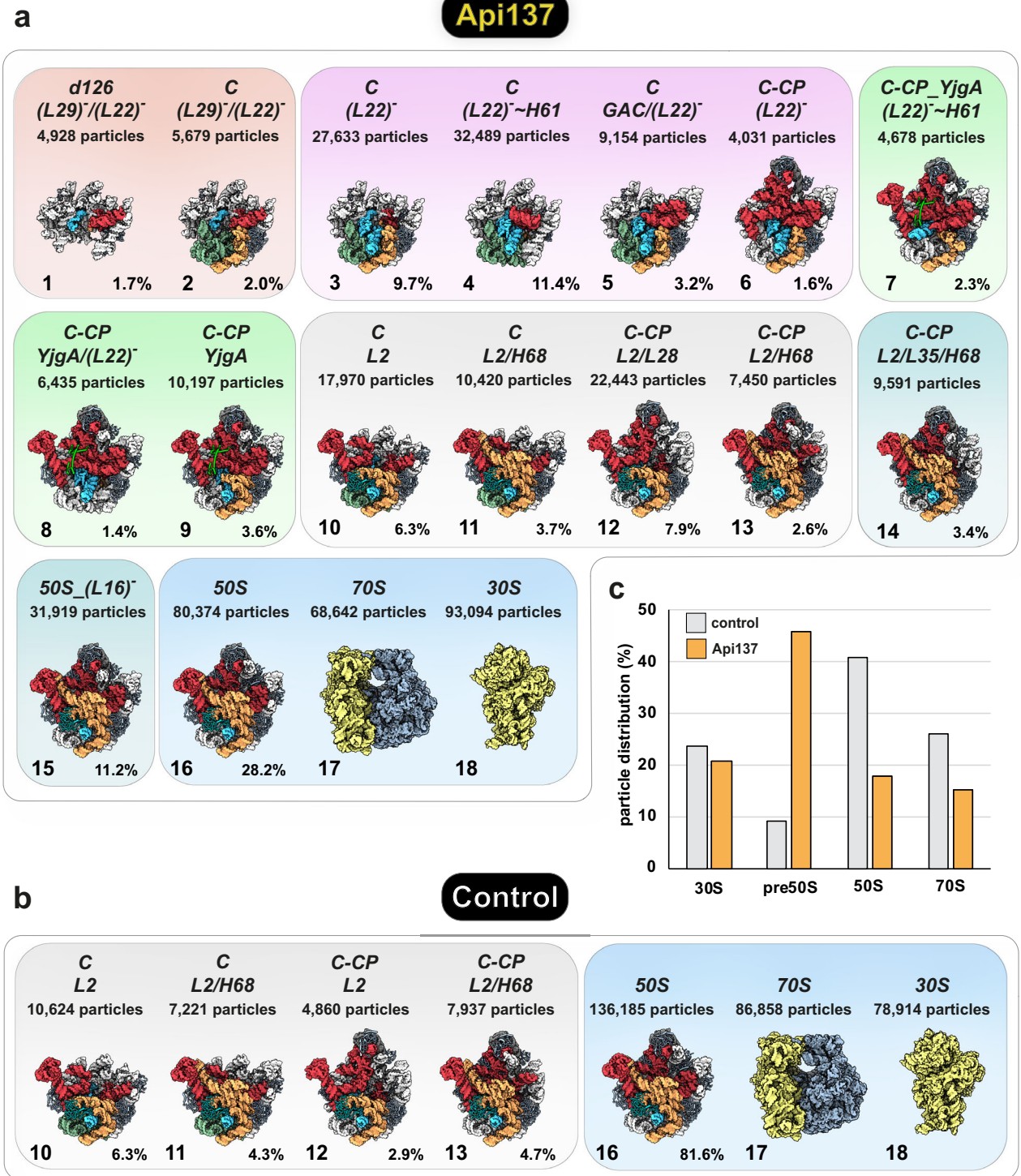

**Fig. 3 | Classes of 70S, 50S, 30S, and pre-50S particles identified in *E. coli* RN31 grown in the presence or the absence of Api137.** States identified in (**a**) the presence of Api137 and (**b**) in the absence of Api137. States are named and color-coded according to structural features and assembly checkpoints. Colored boxes indicate grouping of states based on similar structural characteristics. States are shown as structural surface models at 5 Å resolution. Invariant parts between the pre-50S and 50S states are shown in light gray. Variant parts are color-coded according to the 23S rRNA domain architecture (domain II: cyan, domain IV: dark yellow, domain V: red, 5S rRNA: dark gray). **c** Distribution of 70S ribosomes, 30S and 50S subunits, and pre-50S precursors in both data sets. Light gray: Control, yellow: Api137. Nomenclature: *d*, 23S rRNA domains; C, 50S core; CP, central protuberance; GAC, GTPase associated center; H68, helix 68; L22, large ribosomal subunit protein uL22.

domain V regions. Comparisons revealed that states 3 (*C_(L22)⁻*), 4 (*C_(L22)⁻ - H61*), and 12 (*C-CP_L2/L28*) were most abundant in the Api137 dataset. These findings corroborate a global effect of Api137 on the assembly of 50S subunits.

**Api137 treatment results in pre-50S precursors lacking the early assembly L-proteins uL22 and uL29**

Most of the unique states identified in the Api137 dataset represent less mature states than those present in the control. To elucidate the

underlying molecular defects leading to the accumulation of precursor particles, the most immature states from the Api137 data set (*d126_(L29)ˉ/(L22)ˉ* & *C_(L22)ˉ*) and the control data set (*C_L2*) were compared (Fig. 4a–c). While the *d126_(L29)ˉ/(L22)ˉ* precursor exhibits structured domains I, II, and VI, it lacks cryo-EM density for domain III of the 23S rRNA (Fig. 4d). Formation of domain III coincides with the presence of uL29 and uL23, as observed in a previous study[35]. The absence of the two L-proteins uL22 and uL29 in *d126_(L29)ˉ/(L22)ˉ* is surprising since both of which had been found incorporated in the so-far earliest described precursor of the 50S subunit, *d1*, interacting directly with 23S rRNA domain I[35,49] (Supplementary Fig. 7). Apparently, the absence of uL22 and uL29 impedes structural stabilization of domain III, which represents the next step towards core formation during 50S assembly.

This significant deviation from previous in vitro assembly studies prompted us to investigate the presence of these L-proteins across all identified states. Notably, another precursor, *C_(L29)ˉ/(L22)ˉ*, displayed a complete core but lacked densities for uL22, uL29, bL23 (positioned adjacent to uL29), and bL32 (positioned adjacent to uL22), indicating that domain III can form and is at least partially stabilized even without uL22 and uL29. All other precursors contained uL29 and bL23, suggesting a potentially delayed but eventual incorporation of these L-proteins. Importantly, the absence of uL22 was consistently observed in most identified precursors (states: 1-8), implying a systematic effect of Api137 on ribosome assembly.

### Absence of early assembly L-protein uL22 results in misfolded pre-50S particles

Unexpectedly, the *C_(L22)ˉ - H61* precursor showed a continuous, well-defined density at the expected uL22 binding site, corresponding to RNA rather than protein (Fig. 5a–c). In contrast to most other precursors, H63 of domain IV was not docked in its terminal position. Tentative modeling revealed a non-canonical turn within the root of domain IV (Fig. 5b, c), causing H61 to thread through the gap created by the absence of both uL22 and bL32 (Fig. 5d). As a result, the entire domain IV is positioned on the solvent side (back) of the subunit, preventing the docking of the respective helices in their correct positions, which renders this particle a potential dead-end state. Despite this massive misorientation of H61 and the entire domain IV, *C_(L22)ˉ - H61* showed structurally mature domain V helices H90-H93 located next to the PTC. Since the binding of uL2 always required the previous incorporation of uL22 and bL32, the binding of uL2 may reflect a checkpoint in late 50S assembly (Supplementary Fig. 8). Quantification of uL22 and uL29 incorporation revealed a frequent absence of uL22, observed in ~46.4% of all precursors. Additional misorientation of domain IV regions was observed in nearly half of these cases, resulting in an overall frequency of 18.1% for the *-H61* states. In contrast, the absence of uL29 was observed only in the earliest precursor states, representing 5.1% of all precursors (Fig. 5e).

### Identification of the biogenesis factor YjgA in Api137-induced 50S precursors

Remarkably, three intermediate states were identified that stably bound the biogenesis factor YjgA[50] and showed complete maturation of the domain V helices, including the helices H90 to H93 (Fig. 6), which are part of the subunit´s FC. In state 7 (*C-CP_YjgA_(L22)ˉ - H61*), the domain IV helices were not visible in their expected locations. Additionally, uL22 and bL32 were absent, consistent with the domain IV root's misfolding and the misorientation of H61 towards the back of the subunit (Fig. 6a, b). State 8 (*C-CP_YjgA_(L22)ˉ*) also lacked uL22 and bL32 and exhibited an alternative RNA meshwork at the interface side of the subunit. Notably, H63 of domain IV was correctly docked, but the preceding helices H61 and H62 displayed non-canonical positions. H61 was shifted and tilted toward the center of the subunit, while the base of H62 appeared to be involved in aberrant contacts with uL14. As

a result, H34, which serves as the base for several domain IV helices within the mature 50S subunit, was bent towards and appeared to interact with YjgA (Fig. 6c). The terminal tip of H93 was in contact with H34 and showed a slight downward shift. In state 9 (*C-CP_YjgA*), incorporation of uL22 and bL32 was observed, with helices H61 to H63 positioned in their mature conformations (Fig. 6d).

### Absence of early L-proteins reroutes 50S assembly

Even the correct assembly and folding of the domain IV root appeared to delay the maturation of the following domain IV helices (Supplementary Fig. 9). In the states from the control dataset, we consistently observed mature helices H61 to H67 of domain IV. However, in most precursors specific to the Api137 dataset, a less mature state of domain IV was observed. The delayed maturation of domain IV appeared to coincide with a temporal reorganization of the assembly process, as domain V helices, especially helices H90 to H93 of the FC, were often found to be fully formed (Fig. 4, Supplementary Fig. 9). Additionally, although the total number of precursors without a formed CP increased, the relative proportion between the control and Api137 datasets remained similar. Notably, the region of domain V corresponding to H76 - H78, which form the L1 stalk, was also less frequently matured in precursors of the Api137 dataset. Furthermore, domain II helix H34, which serves as the foundation for positioning domain IV helices H64 to H67 during subsequent assembly, was consistently docked in all but two precursors (states *d126_(L29)ˉ/(L22)ˉ* & *C_(L29)ˉ/(L22)ˉ*). As the maturation of these domain II and IV regions was previously observed to coincide[35], the accumulation of these precursors further suggests a delayed maturation of domain IV. The absence of helix H34 was observed only in precursors that also lacked uL29 and bL23, indicating that these proteins are necessary for the proper formation of these critical regions in domains II and IV regions. The absence of uL22 and bL32 in several states of the Api137 dataset also led to the absence of uL2 in eight precursor states (states 1-8), suggesting a possible role of uL2 as a checkpoint protein, whose binding completes a construction phase.

### Api137 binds to purified precursors

Recently, we identified a second binding site for Api137 on the 50S, located at the PET exit pore in close proximity to the ribosomal proteins uL22 and uL29[33]. Since none of the precursors, 50S or 70S ribosomes, revealed a defined density for Api137, either at its canonical binding site within the PET or at its binding site at the PET exit pore, we considered the possibility that it had dissociated during sample preparation, possibly during the sucrose gradient purification step. To test this hypothesis and provide direct evidence of Api137's ability to bind to the precursors, they were incubated for 20 minutes with 30 μmol/L Api137, representing a 100-fold excess relative to the ribosomal fraction. Precursors were plunge-frozen, and a high-resolution cryo-EM dataset was acquired and processed (Supplementary Fig. 10 & 11).

Now, the 50S class showed a clear extra density for Api137 at the canonical PET binding site (Supplementary Fig. 12e) and a defined extra density at the PET exit pore site, corresponding to a second Api137 molecule, as observed previously[33]. However, even late precursors did not show Api137 density at the PET binding site within the tunnel (Supplementary Fig. 12d)[14,33], most probably because the PTC matures last during 50S assembly[34,36]. In contrast, the second Api137 binding site at the PET exit pore site was found occupied in the precursor states. After class identification, states were grouped into *d126_(L22)ˉ/(L29)ˉ*, *C_(L22)ˉ*, *C_(L22)ˉ - H61*, *C_L22*, *C-CP_(L22)ˉ*, *C-CP_(L22)ˉ - H61*, and *C-CP_L22* classes, 50S and 70S classes, and pooled again to increase particle numbers with a specific organization at the PET exit pore site (Fig. 7). An additional focused classification step was performed on the pooled classes with a fragmented or absent density for uL22, using a local mask on the uL22, Api137, and uL29 binding sites to ensure that uL22 was absent in all particles of this subset. Atomic models of Api137 within the PET and at the PET exit site from our

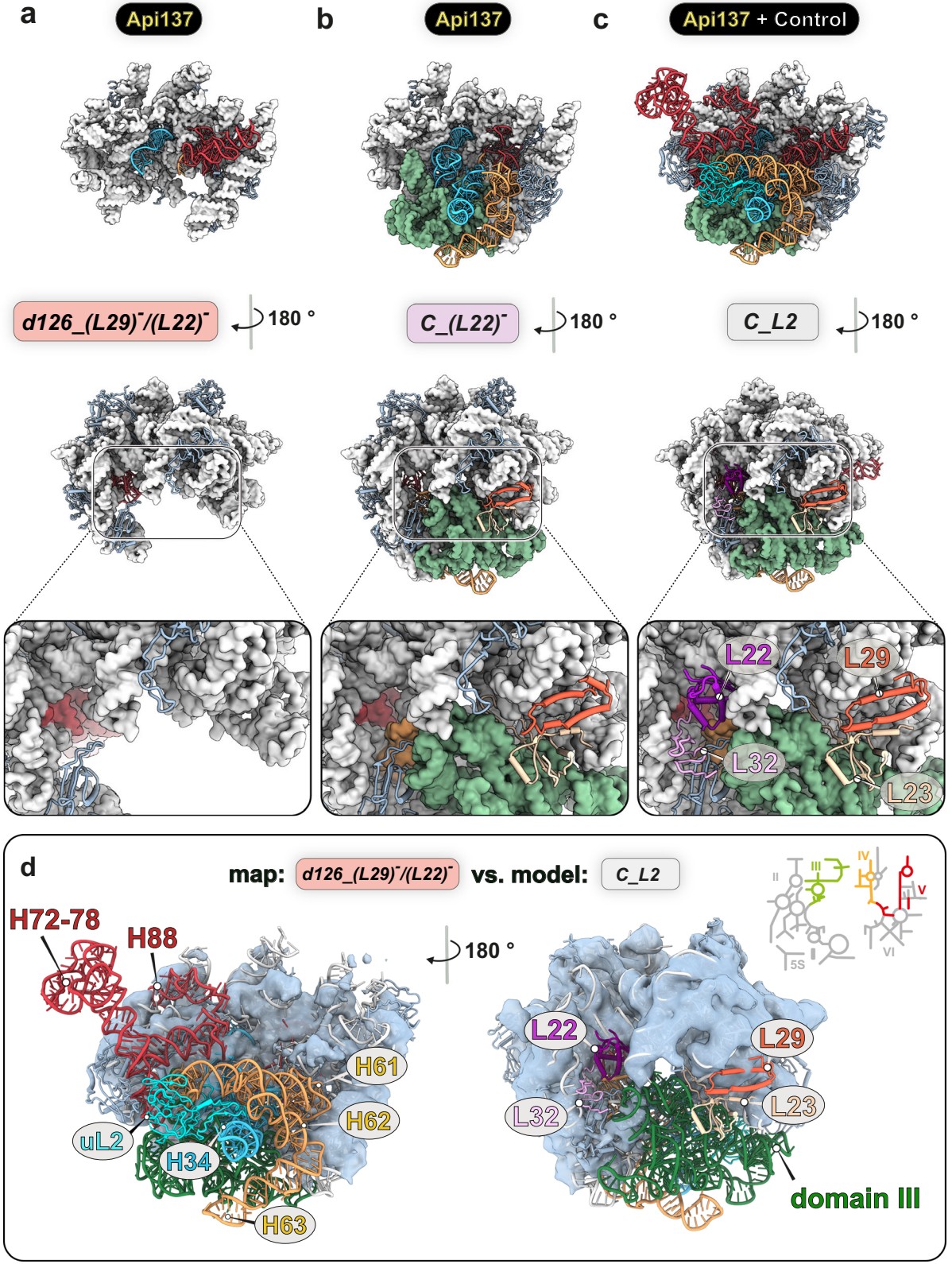

**Fig. 4 | Api137 treatment results in the absence of the early-binding L-proteins uL22 and uL29 in pre-50S precursors. a–c** States *d126_(L29)⁻/(L22)⁻*, *C_(L22)⁻*, and *C_L2* shown in crown view and rotated by 180°. Invariant regions of the 23S rRNA appear as a light gray structural surface model at 5 Å resolution. Variant regions between the pre-50S precursors are color-coded. Variant regions of the 23S rRNA of domains II (light blue), domains IV (yellow), and V (red) are shown as ribbons, and III (green) as a structural surface model. Close-ups show uL22 and uL29 binding sites at the solvent side (back) of the precursors. *(L22)⁻*: Absence of uL22 and bL32. *(L29)⁻/(L22)⁻*: Absence of uL29, uL22, bL32, and bL23. **d** Model of the *C_L2* state identified in the control data set was rigid-body fitted to the map of the *d126_(L29)⁻/(L22)⁻* state identified in the Api137 data set. Absent L-proteins and rRNA segments *d126_(L29)⁻/(L22)⁻* are colored in the model and 2D rRNA diagram.

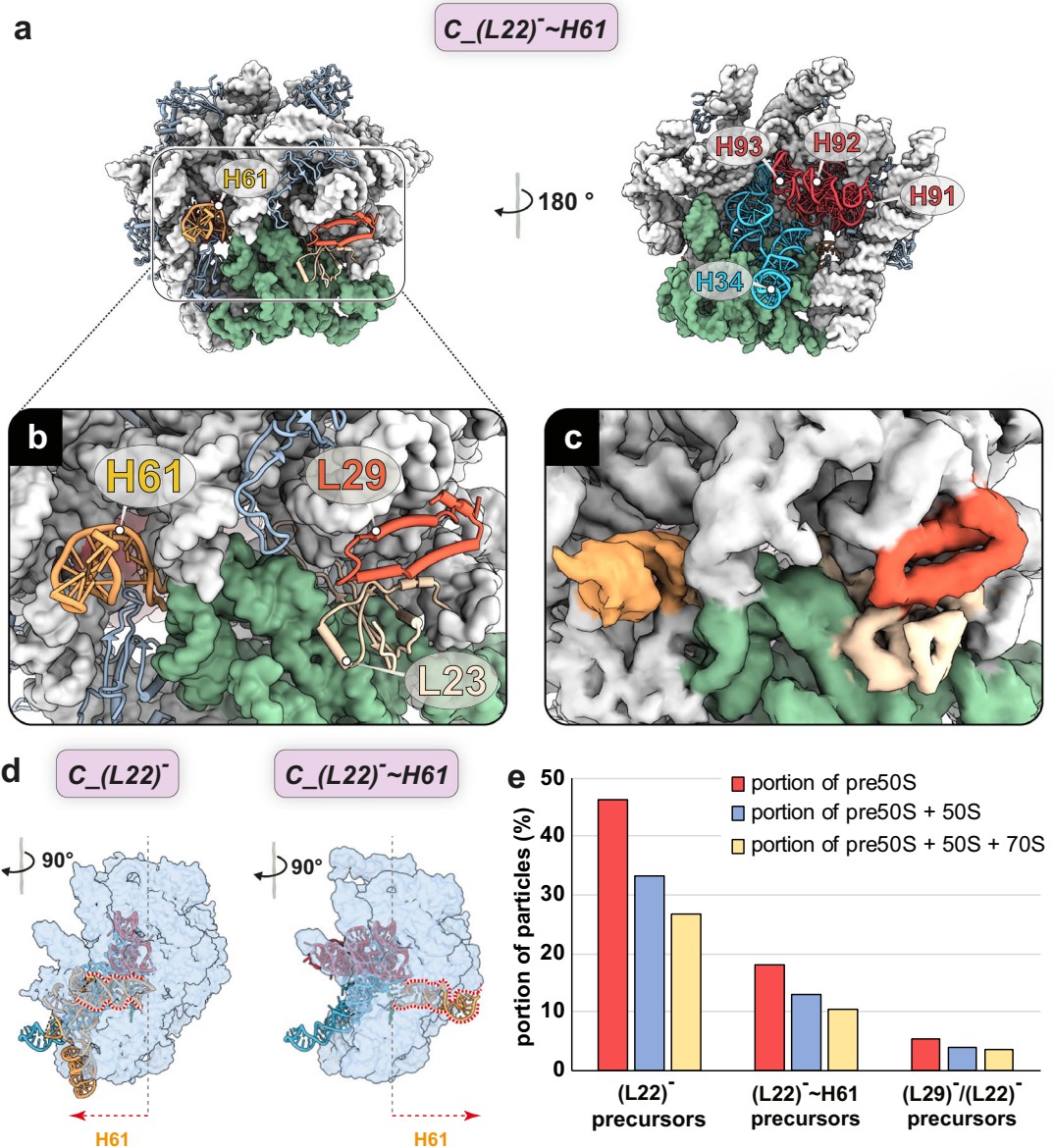

**Fig. 5 | Api137 treatment leads to misorientation of domain IV helices of the 23S rRNA. a** State C_(L22)⁻~H61 is shown in crown view and rotated by 180°. Invariant parts (light gray) and variant domain III (green) are shown as a structural surface model at 5 Å resolution. Variant regions of domains II (light blue), IV (yellow), and V (red) of the 23S rRNA are shown as ribbons. Close-ups with model **b** and local cryo-EM map **c** of the uL22 and uL29 binding sites on the solvent side (back) of the precursors. **d** Trajectory of the base and H61 of domain IV in C_(L22)⁻ and C_(L22)⁻~H61. **e** Percentage of states with indicated assembly defects (absence of L-proteins density or misoriented H61) within the population of pre-50S (red), pre-50S, and 50S (blue) and all particles (yellow). (L22)⁻: absence of uL22 and bL32. C_(L22)⁻~H61: absence of uL22 and bL32 and misfolded domain IV. (L29)⁻/(L22)⁻: absence of uL29, uL22, bL32, and bL23. Fractions are shown for precursors (red), precursors + 50S (blue), and precursors 50S, and 70S (yellow).

previous high resolution 50S structure[33] were rigid-body fitted into each of these pooled states.

Classes that showed both uL22 and uL29 densities exhibited a defined Api137 density at the exit pore binding site[33] (Fig. 7c, Supplementary Fig. 13). In addition, both the (L22)⁻~H61 class and (L22)⁻ class showed a less defined density for Api137, indicating that the peptide can bind even in the absence of uL22 (Fig. 7a, b) but remains more flexible. In the (L22)⁻ states, Api137 showed alternative C-terminal trajectories (Fig. 7d, Supplementary Fig. 13e). Two trajectories could be followed, one of which showed that the peptide projects linearly toward the uL22 binding site (Fig. 7e). Interestingly, this conformation would sterically clash with uL22 if it were present. No density was observed for classes that lacked both the early L-proteins uL22 and uL29, together with domain III.

Taken together these data indicate that Api137 can interact with pre-50S precursors as soon as the PET exit binding site is established. This interaction potentially blocks the incorporation of uL22 and provides a mechanistic explanation for direct inhibition of ribosome assembly by Api137.

## Discussion

Several structural studies focusing on the assembly of the bacterial large ribosomal subunit have revealed the order of events and interdependencies involved, providing a better mechanistic understanding of this process[22,34–36,38,41,46,51]. A key motivation for studying ribosome assembly is to develop molecules that inhibit this process, as ribosome assembly is considered an attractive target for antimicrobial drugs[52,53]. This study revisited earlier findings indicating that *E. coli* cells treated

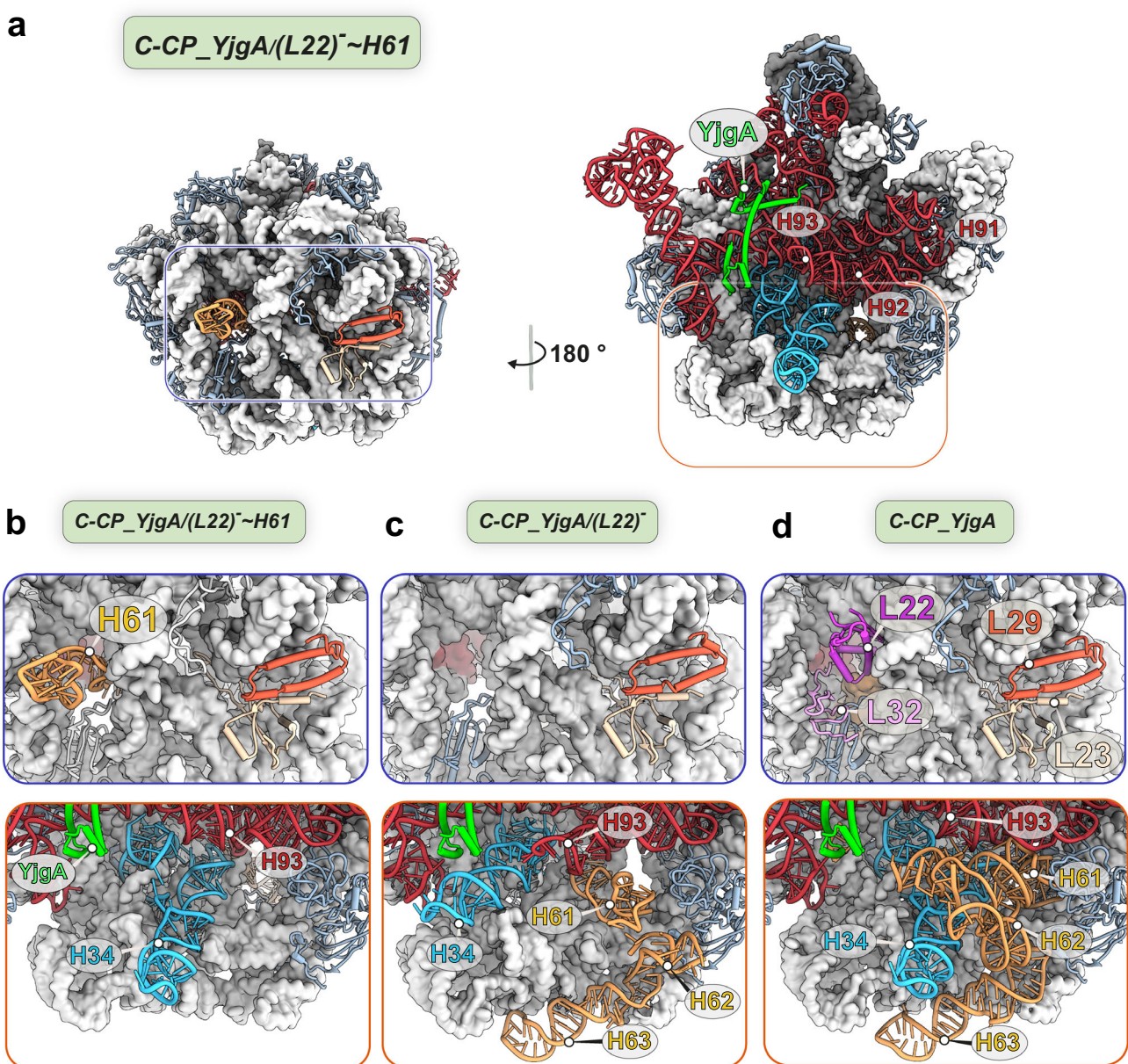

**Fig. 6 | pre-50S precursors with bound biogenesis factor YjgA. a** State *C-CP_YjgA_(L22)⁻ - H61* is shown in crown view and rotated by 180°. Invariant parts are shown as a light gray (23S rRNA) or a dark gray (5S rRNA) structural surface model at 5 Å resolution. Variant parts appear as ribbons and are colored according to the 23S rRNA domain architecture (domain II: cyan, domain V: red). Close-up views show YjgA (green) and the meshwork of rRNA helices at the subunit interfaces. **b–d** Close-up views of the uL22 and uL29 regions at the subunit back (top panels) and interface side (bottom panels) of (**b**) *C-CP_YjgA_(L22)⁻ - H61*, (**c**) *C-CP_YjgA_(L22)⁻*, and (**d**) *C-CP_YjgA*.

with Api137 accumulate incompletely assembled precursors of the 50S ribosomal subunit[13]. Employing a strain that synthesizes ribosomes with distinct fluorescent protein labels for their subunits, we detected increased levels of incompletely assembled 50S ribosomal precursors following Api137 treatment, as determined through sucrose gradient centrifugation of cell lysates and subsequent fluorescence readout. Cryo-EM analysis revealed that various accumulating precursors lack the early-binding L-proteins uL22 and uL29. This deficiency leads to a delayed folding or misorientation of 23S rRNA domain IV, rendering these pre-50S precursors seemingly incapable of maturation. These observations provide initial structural insights into a site-specific drug-induced perturbation of ribosomal assembly.

Tagging the ribosomal proteins bS20 and bL19 with mCherry and EGFP, respectively, did not affect the microbiological properties of *E. coli* RN31 compared to *E. coli* MC4100, at least for Api137 and Onc112,

which was surprising because the tagged proteins could reduce the dynamic interactions of other proteins binding to the ribosome or lead to new interactions affecting ribosome activity. Even the growth rates of both *E. coli* strains in the exponential growth phase were equally suppressed. Considering that the $OD_{600}$ value of the cells incubated with Api137 doubled within 90 min, this indicates that the number of cells doubled, and most likely, the number of ribosomes also doubled. Half of the ribosomes are most likely formed during the 90-minute growth phase since ribosomes are long-lived ribonucleoprotein complexes that are not degraded during the exponential growth rate[54]. Therefore, a more or less stable number of ribosomes per cell can be assumed[55], although the presence of Api137 may affect this ratio. In this context, the significant 3-fold increase of the peak area in the pre-50S region, which corresponded to ~38% of the total EGFP fluorescence in the 70S to pre-50S region compared to ~13% in the control (Fig. 2), and

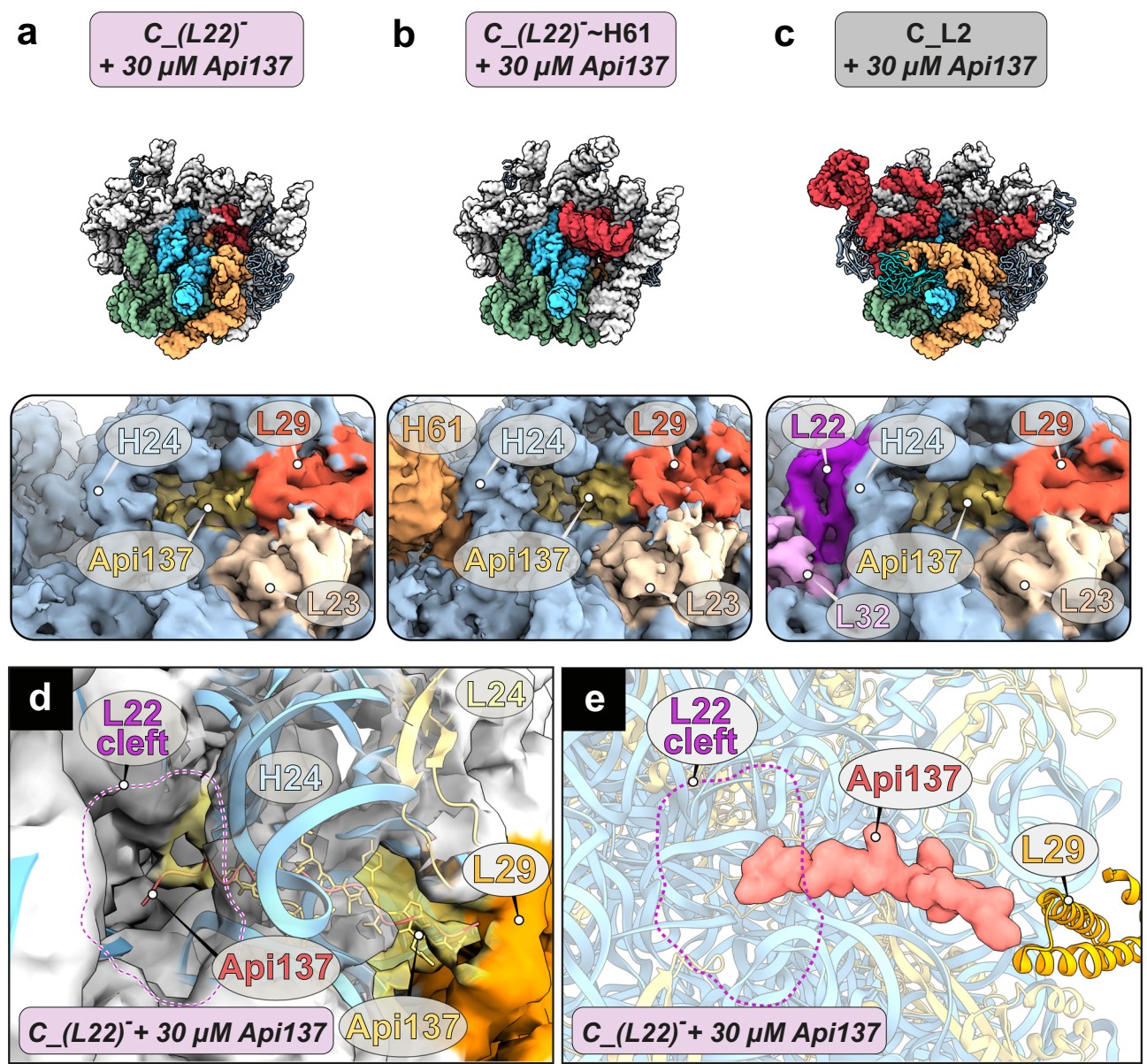

**Fig. 7 | Structures of pre-50S intermediates with cryo-EM density for Api137.** The binding of Api137 (yellow) to the intermediates (**a**) *C_(L22)⁻*, (**b**) *C_(L22)⁻ - H61*, and (**c**) the canonical 50S precursor *C_L2*. States are shown as structural surface models at 5 Å resolution. Invariant regions are colored in light gray. Variant parts are color-coded according to the 23S rRNA domain architecture (domain II: cyan, domain IV: dark yellow, domain V: red, 5S rRNA: dark gray). The peptide binding site is in close proximity to uL29 (orange), uL23 (light brown), uL22 (purple), and L32 (pink). **d**, **e** In *C_(L22)⁻* Api137 (red) adopts a different conformation than in the mature 50S (yellow).

especially the cryo-EM-based observation that 45.9% of the assigned particles in the 50S and pre-50S region resemble misfolded or immature states suggest that the assembly of new 50S subunits was largely disturbed during bacterial growth.

Taken together, in the presence of Api137, cells grow slower and accumulate pre-50S precursors, likely directly mediated by the presence of Api137 in the cell. Some of these accumulating precursors may not mature properly, reducing the productivity of the translational machinery, which ultimately can result in cytotoxic effects.

Cumulative evidence from cryo-EM studies utilizing perturbations such as knockouts of assembly factors or ribosomal proteins, along with previous studies using an in vitro system for reconstitution of the 50S subunit, has revealed that domains I, II, III, and VI of the 23S rRNA fold early in the assembly process to form the 50S subunit´s core[34,36,38–41,56]. In addition, a more refined time-resolved study has elucidated that although assembly could potentially initiate at any

position within the 23S rRNA in the in vitro system, it specifically begins with domain I[35]. In its continuation, 50S assembly can take different routes and concludes its early phase upon the structural completion of the subunit´s core[35,46].

Treatment of *E. coli* cells with Api137 disrupts 50S subunit assembly, leading to the accumulation of precursors lacking density for uL22, akin to an uL22 knockout. This disruption results in the accumulation of previously undescribed novel intermediates, providing deeper insights into the intricacies of 50S assembly.

Particles derived from non-treated cells display the expected maturation pathway (canonical route) (Fig. 8a), with a core particle (state 10) being the most immature state identified that upon CP-formation, folding of H68 and bL35 incorporation matures to a *C-CP* state (state 13) that has a still incompletely folded FC (with disordered helices H90-93) and awaits incorporation of uL16 and final folding of the FC, which ultimately activates the 50S subunit[34,35].

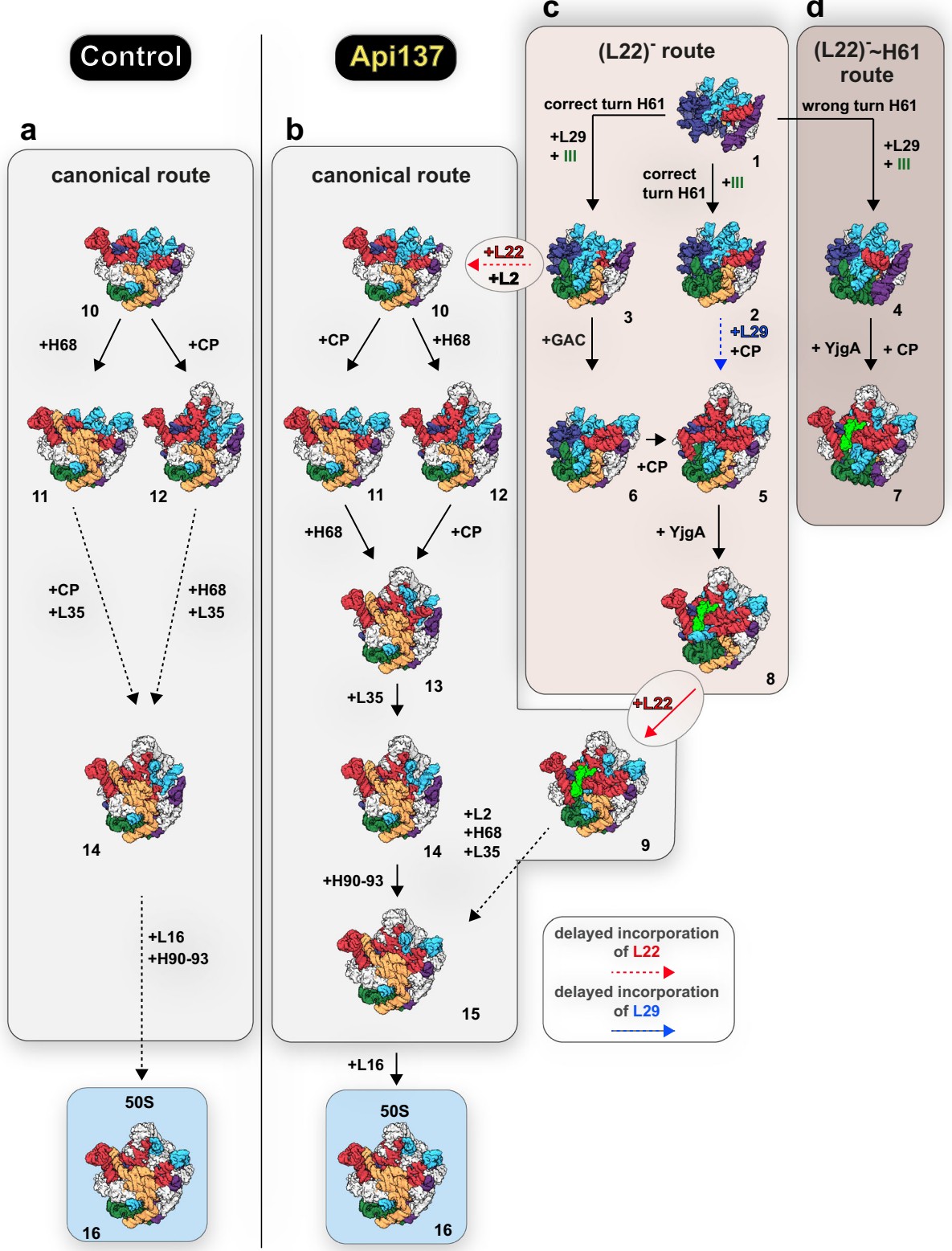

**Fig. 8 | Folding trajectories of pre-50S precursors. a** Intermediates identified in unperturbed control cells (canonical route). **b**–**d** Accumulating precursors upon Api137 treatment corresponding to the canonical route (**b**), or representing alternative folding pathways due to the absence of the early L-proteins uL29, uL22, uL23, and bL32 (**c**, **d**). State 1 is characterized by the absence of all four L-proteins, resulting in either the correct formation of domain IV helices due to correct orientation of H61 (**c**, *(L22)⁻ route*), or accumulation of potential dead-end

precursors (**d**, *(L22)⁻ - H61 route*). Precursors following this pathway have prematurely folded domain V helices (e.g., H90 to H93), whereas canonical precursors (light gray) in control and Api137 samples fold domain IV helices only when uL22 is incorporated. The transition between the two perturbed pathways (**c**, **d**) and the canonical pathway (**b**) can occur via delayed incorporation of uL22/ bL32 and uL29/ uL23.

Analysis of the Api137 dataset revealed additional structural states. These include states (Fig. 8b) that correspond to the canonical route (states 10, 11, 12 and 14) plus two almost mature intermediates (states 13 and 15), mature 50S, and an intermediate with density for the assembly factor YjgA (state 9). This precursor 9 represents a possible connection to a series of particles that appear to mature along a non-canonical route, which comprises two corridors. One of them, the (L22)⁻ route (Fig. 8c), starts with the unusual early state 1, lacking density for uL22, while the second route, branching from this state, seems to yield two states incapable of further maturation (states 4 and 7). These two states share a very unusual feature never observed in any 50S assembly study. The orientation of H61 is turned by 180°, resulting in a progression towards the solvent side instead of the interface side (Fig. 5d). State 7 exhibits a matured CP and domain V. However, the assembly of domain IV is not accomplished, and the uL2 binding site is consequently absent, preventing uL2 binding at this stage. While states in the (L22)⁻ - H61 route (Fig. 8d) do not appear to have the opportunity to transition to the canonical route, states within the (L22)⁻ route, such as states 3 and 8, can do so. The incorporation of uL22 and uL2 converts state 3 to state 10, and the incorporation of uL22 converts state 8 to state 9, allowing further maturation along the canonical route. The absence of uL22 has never been observed neither in bacterial nor eukaryotic LSU intermediates. The structural implications are enormous, with partial misfolding and delayed folding of domain IV being the primary consequences. In this context, it is noteworthy that states, even though lacking uL22 with the associated delay in domain IV folding, can advance and acquire the CP, fold the L1 stalk, i.e., develop nearly all structural features (in particular folding of domain V) without having finished domain IV (states 7-9). In all these cases, the assembly factor YjgA is found to be associated with the particles and interacts with domain V, as described previously. Davis et al. observed YjgA attached to pre-50S precursors accumulating in a bL17 deletion mutant (suffering from domain III assembly defects)[36], while Sheng and colleagues found similar YjgA-bound pre-50S precursors when depleting the RNA helicase DeaD[41]. Based on our structural data, it appears as if YjgA attached to a pre-50S precursor with an already finished domain V awaits domain IV to be completed. However, the subsequent formation of H68 and the binding of uL2 are still consistent with the presence of YjgA, as shown by Nikolay et al. in a bona fide pre-50S precursor. This process is accompanied by conformational changes within YjgA, reinforcing its dynamic contribution to domain IV and V-related late folding steps of the 50S subunit[22]. In addition, Du and colleagues speculated that the flexible, negatively charged N-terminus of YjgA orchestrates late assembly events near the PTC[57].

Initially, we did not observe a cryo-EM density for Api137 either in the precursor states or in the mature 50S subunit. The major binding site of Api137 is located within the polypeptide exit tunnel (PET) next to the PTC of the 50S subunit[14,33]. In most pre-50S precursors identified in this study, however, this binding site has not yet formed as it is one of the final steps in the assembly process[34–36]. A second binding site previously described at the exit pore of the PET[33] is already present in early precursors due to the early folding of rRNA helices H6, H7, and H24 (Supplementary Fig. 6b, c). Supplementing Api137 to the purified precursors revealed that Api137 only binds to the PTC binding site in mature 50S subunits. In contrast, the peptide was detected in most precursors at the recently discovered PET exit binding site. Interestingly, this site is in close proximity to the docking sites of the L-proteins uL22 (C-terminal region of Api137) and uL29 (N-terminal region). Since the incorporation of other L-proteins appears to be unaffected, it is possible that the presence of the peptide sterically hinders the incorporation of these proteins.

Ribosome assembly defects are well known to accumulate upon treatment with ribosome-directed antibiotics, such as chloramphenicol, erythromycin, and many others[58,59]. The discussion about possible direct effects of these drugs, fueled by the controversial work

of Scott Champney[60,61], was countered by the findings of Siibak et al., who presented data suggesting that the observed ribosome assembly defects are a secondary effect of a general dysregulation of translation[62]. While we cannot rule out a potential secondary effect due to translation inhibition by RF-trapping, several lines of evidence support a direct steric effect of Api137 on the assembling 50S subunit. For instance, Siibak et al. found an equal inhibitory effect on 50S and 30S biogenesis. In contrast, Api137-treated E. coli cells accumulated structurally defined pre-50S particles lacking density for the ribosomal proteins uL22, uL29, or uL16, suggesting a specific and direct effect on early and late stages of large subunit assembly. Our experiment with supplemented Api137 demonstrates that Api137 can also occupy the exit pore binding site in precursors (Fig. 7a, b and d, e). The fragmented cryo-EM density in states C_(L22)⁻ and C_(L22)⁻ - H61 suggests that the presence of Api137 may interfere with the binding of uL22 and uL29 (Fig. 7d, e), potentially explaining the defects observed in the L22⁻ route (Fig. 8c). We also observed a variety of precursors that subsequently follow a previously unknown alternative trajectory, explaining the diversity of ribosomal precursors observed in the sucrose gradient profiles.

The L-proteins uL22, uL29, and uL16 are all encoded by the S10 operon[62,63], which has the gene order uS10, uL3, uL4, uL23, uL2, S19, uL22, S3, uL16, uL29, and uS17 with stop codons and start codons overlapping between uL4 and uL23, uL16 and uL29, and additionally uL29 and uS17. As all proteins of the S10 operon, including S3 and uS17, except uL22, uL16, and uL29, were incorporated into all 50S, 30S, and precursor states, this strongly suggests that Api137 directly interferes with ribosomal assembly rather than attenuating the expression of some ribosomal proteins. Furthermore, the mRNA sequences of uL4, uL22, and bL36 end with the UGA stop codon recognized only by RF2, while the coding sequences for all other proteins end with the UAA stop codon recognized by both RF1 and RF2[64]. Thus, depletion of RF1, RF2, or both in the cell would similarly reduce the release of all ribosomal proteins encoded by the S10 operon or at least uL4 and uL23 together with uL22 in the case of a depleted RF2 pool. This conclusion is further supported by a recent proteomics study performed on Api137-treated cells, which detected equal or even increased levels of uL22[65]. Moreover, the fact that multiple binding sites with different binding modes on the ribosome have been previously observed for PrAMPs[33] indicates that these cationic peptides predominantly interact with the negatively charged rRNA. Arginine-rich peptides can displace RNA- and DNA-binding proteins[66], an effect that could also apply to Api137, which consists of one ornithine and four arginine residues. It is tempting to hypothesize that the combined inhibition of translation and assembly synergistically disrupts the translational activity of bacterial cells at different stages.

Since Api137 targets the highly conserved bacterial ribosome at multiple stages by interfering with both, its assembly and with termination of translation, the development of resistance due to mutations in ribosomal proteins and rRNA is supposed to be a low probability scenario. However, the extent to which each mechanism contributes to the observed antibacterial activity remains to be determined. These mechanisms include uptake and intracellular peptide distribution, and off-target binding, for instance, to DnaK[10,67]. Previous studies by the Mankin lab identified mutations in the 23S rRNA, ribosomal protein uL16, and RF1/RF2 of engineered B and K12 E. coli strains that rendered them less susceptible or partially resistant to Api137 and Api137 derived analogs, with the latter mostly related to mutations in RF[14,68,69]. Skowron et al. reported one promising apidaecin derivative with multiple non-canonical amino acid substitutions able to circumvent the mentioned RF-dependent resistance[69], which leaves room for further studies in the assembly context. In addition, mutations in the sbmA gene, which codes for the peptide transporter utilized by apidaecins, are frequently observed[14]. These sbmA mutations represent a defense mechanism that renders all cellular targets obsolete. To

counteract this cellular reaction, Api variants with higher uptake rates (such as Api88) and more pronounced inhibitory effects towards ribosome assembly are of interest. We are currently pursuing strategies in this direction to develop Api derivatives with higher cytotoxic efficacy, surpassing the development of cellular resistance.

# Methods

## Reagents were obtained from the following companies

AppliChem GmbH (Darmstadt, Germany): 4-(2-Hydroxyethyl)-1-piperazineethanesulfonic acid (HEPES, 99.5%), tris(hydroxymethyl)aminomethane (TRIS); Biosolve BV (Valkenswaard, Netherlands): Acetonitrile (ULC/MS grade), formic acid (≥99%), N,N-dimethylformamide (DMF, >99.8%), and piperidine (>9.5%); Carl Roth GmbH & Co. (Karlsruhe, Germany): Agar-agar (Kobe I), dichloromethane (DCM, 99%), glycerin (99.9%), lysogeny broth (LB) Miller, lysozyme, nutrient broth, silica beats (zirconia silica, 100 µm diameter), sodium dodecyl sulfate (SDS), and trifluoroacetic acid (TFA, ≥99.9%, peptide synthesis grade); Honeywell FlukaTM (Seelze, Germany): Ammonium bicarbonate (99.5%), 1,3-disopropyl-carbodiimide (DIC, ≥98.0%), 1,2-ethandithiole (≥98%), N-ethyldiisopropylamin (DIPEA, 98%), thioanisole (≥98%); Iris Biotech GmbH (Marktredwitz, Germany): 2-(1H-benzotriazol-1-yl)-1,1,3,3-tetramethyluronium hexafluorophosphate (HBTU), D-biotin; Merck Millipore (Darmstadt, Germany): Casein (≥95%), tryptic soy broth (TSB); Riedel-de Haën (Seelze, Germany): bromophenol blue. SERVA (Heidelberg, Germany): Acrylamide/Bis solution (37.5:1), ammonium persulfate (APS), BlueBlock PF buffer, Coomassie brilliant blue G-250, tetramethylethylenediamine (TEMED, 99.9%), trypsin, Tween20®; Sigma Aldrich (Steinheim, Germany): ammonium chloride ($NH_4Cl$, ≥99.8%), 4-benzoyl-L-phenylalanin (BPA, 98%), 5(6)-carboxyfluorescein (Cf, for fluorescence), casein, m-cresol (98%), hydrochloric acid (HCl), 1-hydroxy-benzotriazole hydrate (HOBt, ≥97.0%), magnesium chloride ($MgCl_2$, ≥99%), 2-mercaptoethanol (2-ME, ≥99%), N-methylmorpholine (NMM, 99.5%), Mueller-Hinton broth 2 (MHB2, for microbiology, cation-adjusted), potassium dihydrogen phosphate ($KH_2PO_4$, 99.5%), potassium hydroxide (KOH, 90%), sodium chloride (NaCl, 99.5%), sodium hydrogen phosphate ($Na_2HPO_4$, 99%),TFA (≥99%, HPLC grade), triisopropylsilane (TIS, >98.0%); Thermo Fisher Scientific Inc. (Darmstadt, Germany): DNase I (RNase-free, 1 u/µL); VWR International GmbH (Dresden, Germany): Acetonitrile (≥99.9%), diethyl ether (99.9%), formic acid (FA, ~98%, LC-MS grade).

All 9-fluorenylmethyloxycarbonyl- (Fmoc-) protected amino acid derivatives used for peptide synthesis were obtained from Orpegen Pharma GmbH (Heidelberg, Germany) or Iris Biotech. Water (electrical resistance ≥182 kΩ·m, total organic content <2 ppb) was purified in-house using a PureLab Ultra Analytic system (ELGA Lab Water, Celle, Germany).

The following bacteria were used: *Escherichia coli* MC4100 F⁻ [araD139]B/r DE(argF-lac)169 Lambda⁻ e14⁻ flhD5301 DE(fruK-yeiR) 725(fruA25) relA1 rpsL150(strR) rbsR22 DE(fimB-fimE)632 (::IS1) deoC1) and *E. coli* RN31 (MC4100_L19-EGFP S20-mCherry).

## Peptide synthesis

Peptides Api137 (Gu-ONNRPVYIPRPRPPHPRL-OH; Gu = tetramethyl guanidino, O = L-ornithine), Onc112 (VDKPPYLPRPRPPRrIYNr-NH2, r = D-arginine) were synthesized on solid phase using a multiple synthesizer (SYRO2000, MultiSynTech GmbH, Witten, Germany), Fmoc/ᵗBu chemistry, in situ activation with DIC in the presence of HOBt, and Rink amide or Wang resins to obtain C-terminal peptide amides or acids, respectively[9,24]. The N-terminus of Api137 and Api88 was guanidated with HBTU in the presence of NMM. Peptides were cleaved with TFA containing 12.5% (v/v) scavenger mixture (ethanedithiol, m-cresol, thioanisole, and water; 1:2:2:2 eq.; (v/v/v/v)) and precipitated with ice-cold diethyl ether. All peptides were purified on an Äkta Purifier 10 using a Jupiter $C_{18}$-column (5 µm, 300 Å, 250 x 10 mm, Phenomenex®) with an aqueous acetonitrile gradient in the presence of 0.1% TFA as ion pair reagent. Purity was determined by RP-HPLC using a Jupiter $C_{18}$-column (ID 4.6 mm or 2 mm). Molecular weights were confirmed by matrix-assisted laser desorption/ ionization time-of-flight mass spectrometry (MALDI-TOF-MS; 5800 Proteomic Analyzer; AB Sciex, Darmstadt, Germany) or by ESI-MS (Esquire HCT; Bruker Daltonics, Massachusetts, USA).

## Lambda-red recombineering

Coding sequences of EGFP and mCherry (in combination with kanamycin resistance cassettes (kanR) derived from plasmid pKD4[70] with flanking homologous regions (40–50 nucleotides) for 3´-prime genomic insertion in frame with rplS and rpsT, respectively, were amplified using Phusion DNA polymerase). Polymerase chain reaction (PCR) products of the expected size were purified and electroporated into competent DY330 cells. Successful genomic integration was verified by colony PCR and DNA-sequencing. Genetic modifications were transferred into *E. coli* strain MC4100 by P1-phage transduction and verified by genome sequencing.

## Antimicrobial activity

Minimum inhibitory concentrations (MICs) were determined in triplicate using a liquid broth microdilution assay in sterile 96-well plates (polystyrene F-bottom; ref. 655180, Greiner Bio-One GmbH, Frickenhausen, Germany) with a total volume of 100 µL/well. Aqueous peptide solutions (3 g/L) were serially diluted twofold in 33% TSB medium (9.9 g/L) starting at a concentration of 128 µg/mL in nine steps (50 µL/well). Overnight cultures of bacteria grown in 33% TSB were diluted 30-fold in fresh 33% TSB, and after an incubation period of 4 h (37 °C, 200 rpm), cells were diluted to $1.5 \times 10^7$ cfu/mL based on a McFarland test, and 50 µL were added to each well (final concentration $7.5 \times 10^6$ cfu/mL). Plates were incubated at 37 °C for $20 \pm 2$ h. Absorbance was determined at 600 nm ($OD_{600}$) using a microplate reader (PARADIGMᵀᴹ, Beckman Coulter GmbH, Krefeld, Germany), and the MIC was defined as the lowest peptide concentration preventing visible bacterial growth.

## Bacterial culture

An overnight culture (37 °C, 200 rpm) of *E*. coli MC4100 or RN31 in 33% TSB was used to inoculate fresh 33% TSB media to $OD_{600}$ of 0.05. At $OD_{600} = 0.2$–0.3, the preculture was divided into 50 mL fractions and treated with aqueous peptide or antibiotic solution. Water was added to the control. The bacterial cultures treated with Onc112, Api88, erythromycin, or chloramphenicol were always grown along with cultures treated with water and Api137. The $OD_{600}$ was measured after 90 min of incubation (37 °C, 200 rpm), and the cells were harvested by centrifugation (AllegraᵀᴹM 21 R, rotor C0650, 6000 × g, 5 min, 4 °C). All subsequent steps were performed on ice. The supernatants were discarded and each cell pellet was dissolved in ribosome preparation buffer (5 mL, 20 mmol/L HEPES-KOH, 6 mmol/L $MgCl_2$, 30 mmol/L $NH_4Cl$, 4 mmol/L 2-mercaptoethanol, pH 7.6, 4 °C), transferred to preweighed 15 mL Falconᵀᴹ tubes and centrifuged again (rotor C1015, 6000 × g, 5 min, 4 °C). The supernatants were removed, and the pellets were weighed before storage at -80 °C. The engineered bacterial strain RN31, based on *Escherichia coli* MC4100 F- [araD139]B/r DE(argF-lac) 169 Lambda- e14- flhD5301 DE(fruK-yeiR)725(fruA25) relA1 rpsL150(strR) rbsR22 DE(fimB-fimE)632 (::IS1) deoC1), is available from the Hoffmann lab.

## Lysate preparation

Cell pellets were thawed on ice and suspended in ribosome preparation buffer (5 L/g fresh pellet weight). The suspension was added to a 2 mL Fast-Prep reaction tube (twist cap) containing ~0.4 g silica beads (Carl Roth GmbH + Co. KG, Karlsruhe, Germany, Ø 0,1 mm) and disrupted using Fast-Prep 24TM 5 G (MP Biomedicals, Eschwege, Germany). A total of three disruption cycles (30 s; 6 m/s) were performed

with two-minute breaks on ice between each cycle. The reaction tube was centrifuged (AllegraTM 21 R, F2402H, 5000 × *g*, 4 °C, 5 min), and the supernatant was transferred to a new 1.5 mL Eppendorf tube and centrifuged again (20000 × *g*, 4 °C, 20 min). The supernatant was transferred to a new 1.5 mL Eppendorf tube, and the absorbance was recorded at 260 nm. If not used immediately, the lysate was stored at -80 °C.

## Ribosome Profile Analysis (RPA)

Sucrose solutions (25% and 5%, w/v) were freshly prepared with ribosome preparation buffer, and the 5% sucrose solution (6.5 mL) was transferred to a 13.2 mL centrifuge tube (331372, Beckman Coulter GmbH) and bottomed with the 25% sucrose solution using a cannula. The centrifuge tube was sealed with parafilm to prevent any residual air in the tube and stored at 4 °C, first horizontally for 3 h and then upright for 30 min. Part of the solution (550 μL) at the top of the tube was removed, the bacterial lysate was added (500 μL, $A_{260} = 30$), and centrifuged (Beckman Optima™ L-100 XP, rotor SW 41 Ti rotor (333790), 4 °C, 52500 × *g*, 19 h) without brake. The sucrose solution was transferred from the bottom with a cannula at a flow rate of 1 mL/min using a sample pump P-950 of an ÄKTA Purifier HPLC (Amersham, Freiburg, Germany). The absorbance at 254 nm was measured, and fractions of 150 μL were collected on a 96-well microplate (Greiner Bio-One GmbH, 655209). Fluorescence of EGFP ($\lambda_{ex} = 485$ nm, $\lambda_{em} = 535$ nm) and mCherry ($\lambda_{ex} = 585$ nm, $\lambda_{em} = 635$ nm) was measured using a Paradigm™ (Beckman Coulter) microplate reader.

## 70S ribosome purification

*E. coli* RN31 and MC4100 were cultured (37 °C, 200 rpm) in LB medium and harvested by centrifugation (Avanti, rotor JLA 8.1000, 15900 × *g*, 15 min, 4 °C) at an $OD_{600}$ of 3-4. Cell pellets were washed by suspension in ribosome preparation buffer (4 °C) followed by centrifugation (6000 × *g*, 5 min, 4 °C). Cell pellets were suspended in ribosome preparation buffer (2 mL/g) with freshly added 2-mercapotethanol (4 mmol/L). The cell suspension was disrupted three times using a FastPrep-24™ 5 G instrument (MP Biomedicals, Eschwege, Germany) with BigPrep 50 mL settings (40 s, 4.0 m/s) with 1 min incubation on ice in between. After centrifugation (Allegra, rotor S4180, 1620 × *g*, 5 min, 4 °C), the supernatant was incubated with DNase I (5 U/mL) for 1 h on ice and centrifuged (Avanti, rotor Ti 30.5, 16,000 × *g*, 30 min, 4 °C). Cell debris was removed from the supernatant by two further centrifugations (32,000 × *g*, 60 min, 4 °C) and crude 70S ribosomes were pelleted by centrifugation (Beckman Optima™ LE-80K, rotor Ti 70 (337922), 165,000 × *g*, 17 h, 4 °C). Ribosomes were purified by sucrose cushion ultracentrifugation using standard protocols[71]. Briefly, the pellet was suspended in a mixture of ribosome preparation buffer (12.5 mL), and the same volume of ribosome preparation buffer containing sucrose (1.1 mol/L) was carefully bottomed with a cannula followed by centrifugation (100.000 × *g*, 16 h, 4 °C). If the band pattern on SDS-PAGE indicated non-ribosomal proteins, sucrose cushion ultracentrifugation was repeated. The pellet was resuspended in ribosome preparation buffer (25 ml), centrifuged (100.000 × *g*, 16 h, 4 °C), resuspended in ribosome preparation buffer (-0.1 mL/g) and stored at -80 °C. Ribosome concentration was estimated by $OD_{260}$, assuming 15 AU for a ribosome concentration of 1 g/L[13] and a molecular weight of 2.3 MDa for the 70S ribosome.

## SDS-PAGE

Probes were mixed with 5 × sample buffer (62.5 mmol/L Tris-HCl, pH 6.8, 20% (v/v) glycerol, 2% (w/v) SDS, 5% (v/v) 2-mercaptoethanol, 0.5% (w/v) bromophenol blue), incubated at 95° C for 5 min, and separated by SDS-PAGE (T = 16%, C = 2.67%). Gels were stained with Oriole (Oriole Fluorescent Gel Stain (Bio-Rad Laboratories GmbH, Feldkirchen, Germany)) for 60 min and then stained with Coomassie Brilliant Blue G250[72] overnight.

## Tryptic in-gel digestion

Bands of interest were excised from the gels, cut into small pieces, washed three times with ammonium bicarbonate buffer (50 mmol/L) containing 30% acetonitrile (v/v), washed with acetonitrile for 5 min, and dried. Trypsin (0.1 μg) dissolved in ammonium bicarbonate (20 μL, 3 mmol/L) was added, and the gel pieces were incubated at 37 °C for at least 4 h. The supernatant was either analyzed directly by mass spectrometry or transferred to another tube and combined with the solution obtained by incubating the gel pieces with acetonitrile (20 μL) for 5 min and dried under vacuum (60 °C, 1 h).

## Mass spectrometry

Digests were separated on a nanoACQUITY Ultra Performance LC™ (Waters Corp., Manchester, UK) coupled online to an LTQ Orbitrap XL instrument (Thermo Fisher Scientific, Bremen, Germany). Peptides were loaded in aqueous acetonitrile (3%, v/v) containing formic acid (0.1%, v/v) on a nanoACQUITY UPLC® precolumn (2G-VM Trap 5 μm Symmetry® C18 180 μm x 20 mm column) at a flow rate of 5 μL/min of 99% (v/v) eluent A (water containing 0.1% (v/v) formic acid) and 1% (v/v) eluent B (acetonitrile containing 0.1% (v/v) formic acid). Separation was performed on a nanoACQUITY UPLC® BEH C18 column (100 mm length, internal diameter 75 μm and particle diameter 1.7 μm, 35 °C) at a flow rate of 0.3 μL/min using a gradient consisting of two linear slopes from 1 to 40% eluent B in 18.5 min and from 40–95% eluent B in 5.5 min. The column was equilibrated for 10 min. Samples were ionized using a PepSep Emitter (Bruker Daltonics GmbH & Co. KG, Bremen, Germany) at a spray voltage of 1.8 kV. The temperature of the transfer capillary was set at 200 °C and the voltage of the tube lens at 100 V. A data-dependent acquisition (DDA) approach created with Xcalibur software (version 2.0.7) was used: orbitrap resolution of 60,000 at m/z 400, precursor ion survey scans from m/z 400–2000, tandem mass spectra for the top five most intense signals acquired in the linear ion trap (isolation width 2, activation Q 0.25, normalized collision energy 35%, activation time 30 ms) using dynamic exclusion for 60 s.

## Database search

Raw DDA files were processed with Mascot Distiller (version 2.8.4.0; Matrix Science Ltd, London, UK) to generate mgf peak lists, which were searched with the Mascot search engine (version 2.8.0) using Mascot Daemon (version 2.6.0). The search used the *Escherichia coli* K12 proteome (loaded 07.02.2024 https://www.uniprot.org) complemented with sequences of the GFP and mCherry proteins as a fasta file. Search parameters were: semitrypsin considering two missed cleavage sites, cysteine carbamidomethylation ( + 57.022 Da) as a fixed modification and methionine oxidation ( + 15.995 Da) as variable modifications, peptide tolerance of +/- 10 ppm and MS/MS +/- 0.8 Da. The resulting data files were loaded as a spectral library into Skyline (version 22.2.0.312, https://skyline.ms/project/home/begin.view, MacCoss Lab, Washington, USA) using a score threshold of 0.05 and inclusion of ambiguous matches and using the following Skyline settings: background proteome was the database used for the Mascot search, trypsin(semi) [KR/P] with a maximum of 2 missed cleavages, peptides from 8 to 25 residues in length, carbamidomethyl (C) and oxidation (M) as structural modifications with a maximum of three modifications and one loss for a peptide. For transition settings, the following parameters were selected: "precursor charge 2, 3", "y- and b-fragment ion charge 1, 2", "product ion selection m/z >precursor to 3 ions", "special ions N-terminal to proline", and "auto-select all matching transitions". For the library ion match, tolerance was set to m/z 0.5, ´if a library spectrum is available, pick its most intense ions ´ was selected, and 3 product ions were selected from filtered ion charges and types. Instrument m/z range was set from 50–2000, MS1 filtering for count, Orbitrap (resolving power 60,000 at m/z 400) with acquisition method DDA using only scans within 1 min of MS/MS IDs. Raw mass spectrometry data and database search results are available on the

ProteomeXchange under the data set identifier: PXD051066. Generated spectral libraries and Skyline documents are available at the following link: https://panoramaweb.org/Api137_immature_ribo.url.

### Preparation of Api137 and control samples for cryo-EM

Lysates derived from cells grown in absence (control) and presence of Api137 were subjected to sucrose gradient ultra-centrifugation, as described above. Material from sucrose gradient fractions 44 to 63 (corresponding to the pre-50S region of the gradient) of both runs was subjected to ultrafiltration and adjusted in Tico buffer (20 mmol/L HEPES/KOH, pH 7.6, 30 mmol/L potassium acetate, 6 mmol/L magnesium acetate, 4 mmol/L 2-mercaptoethanol) to a final concentration of 0.3 µmol/L. Moreover, the material from Api137-treated cells was supplemented with an additional 30 µM of Api137 for 20 minutes at 4 °C. Samples (control, Api137-treated and Api137-treated supplemented with further Api137) were applied to glow-discharged (Pelco Easy Glow) holey carbon grids (400 mesh Cu R2/2, Quantifoil MicroTools GmbH) and plunge-frozen in liquid ethane using a Vitrobot Mark IV (Thermo Fisher Scientific) device.

### Cryo-electron microscopy and data processing

Two data sets for precursors from Api137-treated cells were collected on a Titan Krios G3i transmission electron microscope (ThermoFisher Scientific, Server version 2.15.3, TIA version 5.0) operated at an acceleration voltage of 300 kV and equipped with an extra-bright field emission gun, a BioQuantum post-column energy filter (Gatan), and a K3 direct electron detector (Gatan, Digital Micrograph version 3.32.2403.0). All images were acquired in low-dose mode as dose-fractionated movies using EPU version 2.8.1 (ThermoFisher Scientific) with a maximum image shift of 12 µm using aberration-free image shifting.

For the first data set, 2704 movies with a total dose of 45 electrons per Å² (e Å⁻²), each divided into 45 fractions (with an individual dose of 1 e Å⁻² per fraction) were collected in energy-filtered zero-loss (slit width 20 eV) nanoprobe mode at a nominal magnification of × 81.000 (resulting in a calibrated pixel size of 0.53 at the specimen level) in super-resolution mode with a 100 µm objective aperture. Data were recorded for 1.17 s with defocus values ranging from − 0.8 to − 2 µm.

The second data set consisted of 1952 movies with a total dose of 40.7 e Å⁻², each divided into 30 fractions (with an individual dose of 1.35 e Å⁻² per fraction) were recorded in energy-filtered zero-loss (slit width 20 eV), nano-probe mode at a nominal magnification of × 81.000 (resulting in a calibrated pixel size of 0.53 Å at the specimen level) in super-resolution mode with a 100 µm objective aperture. Data were recorded for 0.81 s with defocus values ranging from −0.7 to −2 µm.

A small subset of the first data set (404 micrographs) was pre-processed in CryoSPARC using patch-motion correction (standard parameters) and patch-CTF correction (resolution range: 12 Å - 4 Å; Supplementary Fig. 2). Particles were picked using a blob picker with a particle diameter ranging from 140 to 220 Å. Particles were extracted with a box size of 720 Å and binned on the fly to 180 Å for an initial pixel size of 2.12 Å (Supplementary Fig. 2a). Particles were subjected to one round of multi-class ab initio reconstruction (class similarity = 0) for an initial consensus map, followed by heterogeneous refinement to discard non-ribosomal particles. A consensus refinement was performed on the ribosomal map, followed by a 3D variability analysis. The variability analysis revealed the presence of 70S, 50S, 30S, and pre-50S states structurally similar to the previously identified pre-50S core state.

To improve particle picking, 84 equally spaced projections from each 3D volume (70S, 30S, 50S, pre-50S) were generated in CryoSPARC and filtered to 20 Å, followed by Xmipp3 2D classification (four classes for each set of 84 projections) to generate generalized picking templates. 2D classes were added to one stack for a final set of 16 picking templates. Particles were picked again, now using both

datasets and the previously generated templates in Gautomatch (developed by K. Zhang). Particle images were extracted and normalized using Relion 3.1[73] with a box size of 800, and Fourier cropped to 200 for sorting (Supplementary Fig. 2b).

Newly extracted particles were reassigned to the previously identified 70S, 50S, 30S, and pre-50S states and non-ribosomal classes. Each ribosomal class was refined once, and subclasses were identified using multiple rounds of 3D variability analysis within CryoSPARC (Supplementary Fig. 3a, b). Classes were sorted until no more variability was observed or until particle counts were too low. Several sparsely populated classes still showed well-resolved features which appeared sub-stoichiometric, indicating residual heterogeneity. Hierarchical sorting resulted in a total of 18 classes for the Api137 sample and 8 classes for the control sample. Particles from selected classes were re-extracted using a box size of 800 and Fourier-cropped to a box size of 300 for a final pixel size of 1.41 (Supplementary Fig. 5 and 6). A consensus refinement using on-the-fly local and global CTF refinement was performed using all 50S and pre-50S particles to improve CTF parameters (Supplementary Fig. 3c). Finally, all distinct subclasses were subjected to non-uniform refinement using the improved CTF values. For validation, all final classes were subjected to an ab initio reconstruction.

For the control data set, 3507 movies with a total dose of 45 e Å⁻², each divided into 45 fractions (with an individual dose of 1 e Å⁻² per fraction) were collected in energy-filtered zero-loss (slit width 20 eV) nanoprobe mode at a nominal magnification of × 81.000 (resulting in a calibrated pixel size of 0.53 at the specimen level) in super-resolution mode with a 100 µm objective aperture. Data were recorded for 1.17 s with defocus values ranging from − 0.8 to − 2 µm. This data set was preprocessed the same way as the Api137 data set (Supplementary Fig. 4). In short, particles were picked using previously generated templates for the 30S, pre-50S, 50S, and 70S and extracted using a box size of 800 and Fourier cropped to 200 pixels. Subsequently, extracted particles were subjected to a heterogenous refinement using the ribosomal and non-ribosomal templates that were generated before. 3D variability and subsequent clustering were performed on the ribosomal classes until no more variability was observed. Hierarchical sorting resulted in four pre-50S states, 50S, 30S, and three distinct 70S states.

For the data set of Api137 precursors supplemented with Api137, 2794 movies with a total dose of 46,2 e Å⁻², each divided into 56 fractions (with an individual dose of 1 e Å⁻² per fraction) were collected in energy-filtered zero-loss (slit width 20 eV) nanoprobe mode at a nominal magnification of ×64.000 (resulting in a calibrated pixel size of 0.666 at the specimen level) in super-resolution mode with a 100 µm objective aperture. Data were recorded for 1.2 s with defocus values ranging from − 0.8 to − 2 µm. Generalized picking templates were once again utilized for template-based particle picking with Gautomatch. Particle images were extracted and normalized using Relion 3.1, with a box size of 800 and Fourier-cropped to 200 for sorting. Newly extracted particles from both datasets were reassigned to the previously identified 70S, 50S, 30S, and pre-50S states, as well as to non-ribosomal classes (first reassignment). Subsequently, particles from the 50S, 70S, and pre-50S classes were reassigned to states previously identified in the first Api137 dataset (second reassignment). Multiple non-informative classes were discarded after 2D classification. Ribosomal classes underwent one more round of 3D variability clustering. All ribosomal particles were then reassigned to the newly identified classes (third reassignment). Further variability was addressed through hierarchical 3D variability clustering, resulting in 33 overall classes. To increase particle numbers, classes with the same organization at the PET exit site were pooled again and subjected to a homogenous refinement. Classes with fragmented or absent uL22 density were pooled and subjected to an additional round of focused 3D variability analysis, resulting in a (uL22)⁻ class and a uL22 class.

## Model building and structural analysis

Atomic models of previously identified 50S assembly intermediates[35] were used for modeling of 50S precursors from the Api137 and control data sets (Supplementary Table 1 and 2). Initially, models were rigid-body docked in ChimeraX[74]. RNA regions and L-proteins were removed when densities were missing or strongly fragmented. The adjustment of structured elements was performed by iterative model building and real-space refinement into the EM-density using Coot 0.9.6[75] and Phenix 1.20[76], considering secondary structure restraints. Models for maps with resolutions lower than 4 Å were subjected to a final round of geometry minimization in Phenix for restoration of the geometry of the RNA.

For the analysis of maps of 50S precursors from Api137 treated cells, supplemented with 30 μM Api137, models built for the first data set were rigid-body fitted into the corresponding cryo-EM density maps (Supplementary Table 3). Therefore, models of principal states (C, C-CP or 50S) with a specific organization at the PET exit site ((L22)ʹ/(L29)ʹ, (L22)ʹ, (L22)ʹ ~ H61 or L22) were used and fitted into the pooled refinement maps. As pooled substates now contained structurally heterogenous regions, models were truncated to the invariant parts between the states. Subsequently, the previously published Api137 atomic models (PDB: 8RPY)[33] were rigid-body fitted into the corresponding local densities within the PET and PET exit binding sites.

## Reporting summary

Further information on research design is available in the Nature Portfolio Reporting Summary linked to this article.

## Data availability

Cryo-EM density maps and atomic models of pre-50S precursors from Api137-treated cells generated in this study have been deposited in EMDB and PDB as follows: EMD-51828, 9H3K (d126_(L29)ʹ/(L22)ʹ), EMD-51829, 9H3L (C_(L29)ʹ/(L22)ʹ), EMD-51830, 9H3M (C_(L22)ʹ), EMD-51831, 9H3N (C_(L22)ʹ ~ H61), EMD-51832, 9H3O (C_GAC_(L22)ʹ), EMD-51833, 9H3P (C-CP_(L22)ʹ), EMD-51834, 9H3Q (C_YjgA_(L22)ʹ), EMD-51835, 9H3R (C_YjgA_(L22)ʹ ~ H61), EMD-51836, 9H3S (C_YjgA), EMD-51837, 9H3T (C_L2), EMD-51838, 9H3U (C_L2/H68), EMD-51839, 9H3V (C-CP_L2/L28), EMD-51840, 9H3W (C-CP_L2-H68), EMD-51841, 9H3X (C-CP_L2/L35-H68), EMD-51842, 9H3Y (50S_(L16)ʹ), EMD-51843, 9H3Z (50S), Cryo-EM density maps and truncated atomic models with rigid-body fitted Api137 of pooled pre-50S states from Api137-treated cells supplemented with Api137 generated in this study have been deposited in EMDB and PDB as follows: EMD-51982, 9HAL (Pooled d126_(L29)ʹ/(L22)ʹ_1); EMD-51983, 9HAM (C_(L29)ʹ/(L22)ʹ); EMD-51973, 9HA1 (Pooled C_(L22)ʹ_2 with the canonical PET exit Api137 conformation); EMD-51974, 9HA2 (Pooled C_(L22)ʹ_2 with the alternative PET exit Api137 conformation); EMD-51975, 9HA3 (Pooled C_(L22)ʹ ~ H61_3); EMD-51976, 9HA4 (Pooled C-CP_(L22)ʹ_5); EMD-51979, 9HA7 (Pooled C-CP_(L22)ʹ ~ H61_6); EMD-51977, 9HA5 (Pooled C_L2_4); EMD-51981, 9HAI (Pooled C_CP_7); EMD-51978, 9HA6 (50S); The following cryo-EM density maps are available via Zenodo: control data set: zenodo entry 13939462, individual pre-50S states from the Api137 treated sample supplemented with Api137: zenodo entry 13919082. The protein mass spectrometry data have been deposited to the ProteomeXchange Consortium via the Panorama Public [https://panoramaweb.org/Api137_immature_ribo.url] partner repository with the dataset identifier PXD051066 [http://proteomecentral.proteomexchange.org/cgi/GetDataset?ID=PXD051066]. Source data are provided with this paper.

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

## Acknowledgements

We thank Daniela Volke for assistance on mass spectrometric analysis, Timo Flügel, Daniel Knappe, Yollete Guillén Schlippe, Norbert Sträter, and Franziska Wiechert for intensive scientific discussions. We thank also the Core Facility for cryo-Electron Microscopy (CFcryoEM) of the Charité - Universitätsmedizin Berlin for support in acquisition (and analysis) of the data. The CFcryoEM was supported by the German Research Foundation (DFG) through grant No. INST 335/588-1 FUGG. This work was funded by Bundesministerium für Bildung und Forschung (BMBF 16GW0300 to C.M.T.S. and 16GW0299K to R.H.) and supported by the Deutsche Forschungsgemeinschaft (DFG) through the cluster of excellence "UniSysCat" (under Germany´s Excellence Strategy-EXC2008/1-390540038 to, C.M.T.S.).

## Author contributions

Data analysis: S.M.L., J.G., M.R., A.K., T.S., and R.N.; Experimental design: S.M.L., J.G., M.R., T.S., A.K., T.S., R.N., C.M.T.S., and R.H.; Funding acquisition: R.N., C.M.T.S., and R.H.; Sample preparation: S.M.L., J.G., M.R., and T.S.; Supervision: A.K., R.N., C.M.T.S., and R.H.; Initial draft: S.M.L. and J.G.; Writing: All authors approved the final manuscript.

## Funding

## Competing interests

The authors declare no competing interests.
