## [Transparent Peer Review file · Nature Communications]

The proline-rich antimicrobial peptide Api137 disrupts large ribosomal subunit assembly and induces misfolding

Corresponding Author: Dr Rainer Nikolay

Version 0:

Reviewer comments:

Reviewer #1

(Remarks to the Author)

Lauer and colleagues unveil a novel mode of action for the proline-rich antimicrobial peptide Api137. In addition to binding ribosomes, thereby stalling them at the termination phase, this peptide disrupts the proper assembly of the major subunit of the bacterial ribosome.

Their manuscript illuminates an uncharted pathway of action for Api137, a discovery of significant interest. However, literature and the authors themselves suggest that Api137 may not stand as a lead candidate for new drug development, having been surpassed by other Apidaecin derivatives. In terms of drug discovery in the field of antimicrobials and combating antibiotic resistance, the impact of the work is therefore limited. However, while the paper does not significantly advance the characterization of a molecule soon to undergo clinical trials, it does shed light on a new antimicrobial mechanism that could inspire the design of future molecules.

The writing and presentation of the paper are commendable, with conclusions well-supported by evidence. However, I believe further harmonization and comparison with the results from Skowron et al. 2023, Baliga et al. 2021, and Florin et al. 2019 in the discussion would enhance its impact. Additionally, providing evidence of Api binding to the involved structures would greatly bolster the work, as while the evidence supporting Api's involvement in ribosome synthesis impairment is convincing, the precise mechanism remains elusive and speculative.

From a methodological standpoint, the paper appears solid and well-constructed. However, my limited expertise in cryoEM of the ribosome precludes me from offering a critical evaluation of the entire process.

The main (and maybe only significant) drawback of the paper, in my opinion, is indeed its highly descriptive nature, which demands a deep understanding of ribosome biology for full comprehension and critic analysis. Consequently, the paper as it is, may be only partially accessible to a broad audience. Authors may consider if the paper may benefit from submission to a more specialized journal focused on structural biology.

MINORS:

Fig 1D, the profile of MC4100 show diffused bandshift compared to RN31. Overall the protein profiles are clear and the profiles can be compared easily, but maybe authors may mention this minor point.

line 167. "...with other PrAMPs" should be in my opinion tuned down, as the comparison has been done with Api88 and Onc112. As it is, the sentence sounds bit too generic with respect to what it has been done.

Line 231. Fix the missing reference

Line 263. Fix the missing reference

Reviewer #2

(Remarks to the Author)

The manuscript analyzes the presence, composition, and structures of accumulated assembly intermediates of the 50S ribosomal subunit when *E. coli* growth is inhibited by Api137, a translation-inhibiting antibacterial peptide derived from honey bees. Comparison with two controls (no treatment, treatment with Onc112 - another translation-inhibiting peptide derived from a milkweed bug) suggests that the 50S assembly intermediates observed are specific to Api137 treatment. Single-particle cryo-EM structure analysis and subsequent image classification is used to characterize the different structural states within the 50S assembly intermediate population. Specific changes such as absence/loss of the proteins uL22, uL23, uL29, and bL32 and structural changes in ribosomal RNA (misalignment of helix 61 (H61)) are found, characterized, and interpreted.

The main finding of the work that Api137 treatment results in ribosome assembly problems seems well-supported by the data presented. I do have a few questions and concerns about the results and interpretations in this work:

1. All of the PDB validation reports seem to have 'not for manuscript review' written on them. Are none of the structures deposited in the RCSB databank? This might mean that the oversight provided by the PDB in quality control of the structures is missing at this time.
2. There seem to be 8 ribosome structures previously deposited in the RCSB databank with Api137 bound in the NPET (PDB IDs: 5O2R, 6GXN, 6GXO, 6GXP, 6GWT, 6YSS, 6YST, 6YSU). In all these structures, the Api137 peptide is bound in virtually the same location. The authors refer to another study of their own and indicate that it is accepted for publication, where they have identified another binding site for Api137 (sometimes accompanied by a mention of reference 13, which is a Api137 structure publication from 2017). There are two panels in Supplementary figure 6 that seem to show a second binding site of Api137 but it is not possible to assess the validity/implications of the second site through data presented here.
3. The number of particles in some of the cryo-EM density reconstructed classes is quite small. It is not clear if an intermediate *ab initio* reconstruction step has been used in generating the final cryo-EM density using only the particle images for each class. If not, then it is necessary to do so. This will demonstrate that each class has sufficient particle image views to be reconstructed without a reference bias introduced by provision of a starting structure.
4. The cryo-EM densities reconstructed from the various particle classes have a range of resolutions. To rule out the possibility that certain protein densities are not visible only due to higher threshold values used for display, cartoon/ribbon models should be shown within their corresponding transparent cryo-EM densities, with each cryo-EM density displayed at a common threshold value with no hiding of the dust/noise (perhaps in a supplementary figure?).
5. Were the three cultures (Api137 treated, Onc112 treated, and no treatment) grown simultaneously such that differences due to all other experimental variables would be minimized? Whether so or whether they were temporally distinct experiments should be clarified in the methods section.
6. The S20-mcherry and L19-EGFP fluorescence assessment of pre-50S populations are fine, but these can only assess populations that have these proteins present. This assessment will therefore miss any assembly intermediates without these proteins. This should be stated in the manuscript.
7. Is the EGFP density observed in all the 50S cryo-EM densities that have L19? If so, is the EGFP model fit into its corresponding density?
8. On lines 180-181, where there is a disclaimer about absence of cryo-EM densities, there is the following line: 'However, at this stage of maturation, when it exhibits full density, its presence becomes unambiguous.' It is not clear which stage of maturation is being referred to and why the presence becomes unambiguous.
9. On lines 194-195 (and Fig 3C), the proportion of 30S subunit particles remaining similar between untreated and Api137 treated samples seems to not agree with either the fluorescence data or the expectation that 30S subunits would accumulate in the absence of their combination with active 50S subunits to form 70S ribosomes. Is there an explanation for this discrepancy?
10. It should be made clear that the number of particle images in cryo-EM classes is only a rough estimate of relative populations in the sample. There could be structural states with a preferred orientation that do not separate into 3D classes and do not get correctly counted.
11. The figures with structures are visually appealing but could be made clearer. In Fig 1, it is not immediately apparent whether the structures shown are structural models or cryo-EM densities. Also in Fig. 1, the 5S RNA is mentioned to be color coded as dark grey, but dark grey parts seem to appear in more than one region of the 50S subunit. It is my personal opinion that having a surface representation for one part of an atomic model with a ribbon representation for another makes it less clear to understand what is being shown, especially for a general readership, and using a mix of color and transparency with a consistent representation (especially ribbon) works better.
12. I am not sure whether presence/absence of a specific proteins in 50S assembly intermediate cryo-EM densities or alternate orientation of helices can be considered as determinants of 'irreversible' or 'dead-end' nature for such intermediates. I think such definitive language might be better avoided.
13. Is the Api137 density not seen in any of the fully-formed ribosome structures (e.g. 70S) in the Api137 treatment sample?

14. The authors seem to suggest a direct steric mechanism for Api137 interference with ribosome assembly (line 433). Both the previously known Api137 binding sites and the novel ones mentioned by the authors are in properly assembled ribosomes. If the authors are proposing other unknown or non-specific binding modes of Api137 in unassembled ribosomal RNA, this should be clearly stated.

15. Could the Fig. 2 title be altered to not mention 'ribosome profile' which has a different default interpretation these days. Perhaps something like 'Statistical ribosome sucrose gradient profile analysis ...' instead?

16. Lines 110 and 129 refers to a URL about mass spectrometric data (https://panoramaweb.org/Api137_immature_ribo.url) that seems to be password-protected.

17. In the pdf file I read, lines 231-232 and 263-264 have a '(Error! 231 Reference source not found.' message.

Version 1:

Reviewer comments:

Reviewer #1

(Remarks to the Author)

After revision, the paper has been in my opinion improved and it is now more accessible to a broader community of readers.

Moreover, the new evidences provided by the authors further corroborates their story, and I appreciated the clarification that Api137 is a tool to study mechanisms to inhibit the ribosome.

Although my technical background on ribosome cryo-EM is limited, I understand that the authors considered the indications of referee 2. Comments of Ref2 are not of my concern, but I personally appreciated that authors took advantage of any comment that could help them in improving the paper.

I have no more concerns and I think the paper is ready for publication, if in line with the Editor's decision.

Reviewer #2

(Remarks to the Author)

The additional experiments with 100-fold excess Api137 that show Api137 cryo-EM density in pre-50S intermediates (Fig. 7) are informative and improve the manuscript. The authors have addressed or replied adequately to all my concerns.

Reviewer #3

(Remarks to the Author)

In this revised manuscript, Lauer et al. presents additional insights into the inhibitory mechanism of the antimicrobial designer peptide Api137 on bacterial ribosome using cryo-EM. They found that in addition to the established binding site near the polypeptide exit tunnel, Api137 induces misfolding of ribosomal components to disrupt assembly of the 50S subunit, suggesting a multi-stage interference of ribosome function. Upon revision, the authors provided additional structures to support the conclusion that an alternative binding site at the exit site of PET emerges during assembly, which may explain the incomplete 50S assembly due to steric hinderance with uL22.

The major drawback of tagging ribosomal proteins with fluorescent protein reporters lies in the bulky sizes of EGFP and mCherry that might interfere with assembly and maturation, especially for target proteins of low molecular weights. However, the authors have convincingly demonstrated successful incorporation and persistent accumulation of pre 50S subunits in the presence or absence of fluorescently tagged markers using an array of techniques (mass spec, ribosome profiling, fluorometric profiling). This mechanism is strongly supported by a series of cryo-EM density maps at respectable resolution and further elucidated through structural comparisons across different binding states. Additionally, the interaction between the Api137 ligand and the protein is clearly resolved in the EM map, offering valuable insights into antibiotic resistance and guiding the potential development of antimicrobial drugs.

Overall, I believe the major conclusions are well supported by experimental evidence and the additional data acquired in the first round of revision significantly strengthened the manuscript. I support the publication of this work in Nature Communication.

Reviewer #4

(Remarks to the Author)

Reviewer #1 (Remarks to the Author):

Lauer and colleagues unveil a novel mode of action for the proline-rich antimicrobial peptide Api137. In addition to binding ribosomes, thereby stalling them at the termination phase, this peptide disrupts the proper assembly of the major subunit of the bacterial ribosome.

Their manuscript illuminates an uncharted pathway of action for Api137, a discovery of significant interest.

> We are pleased that reviewer #1 considers the discovery of *an uncharted pathway of action for Api137* to be of significant interest.

However, literature and the authors themselves suggest that Api137 may not stand as a lead candidate for new drug development, having been surpassed by other Apidaecin derivatives.

> It was not our intention to give the impression that Api137 was outperformed by other apidaecin derivatives. Actually, we are not aware of any more active apidaecin derivatives. There are a few analogues with similar activity, but their efficacy has not been tested in animal models and they are presumed to have degradation issues.

In terms of drug discovery in the field of antimicrobials and combating antibiotic resistance, the impact of the work is therefore limited. However, while the paper does not significantly advance the characterization of a molecule soon to undergo clinical trials, it does shed light on a new antimicrobial mechanism that could inspire the design of future molecules.

> The reviewer speaks from our hearts. We added a corresponding sentence to conclude our introduction:

Lines 97-99 “The finding that antimicrobial molecules, such as apidaecin peptides, can induce misfolding of ribosomal components during assembly offers an attractive avenue for future research and potential drug development.”

1.1 The writing and presentation of the paper are commendable, with conclusions well-supported by evidence. However, I believe further harmonization and comparison with the results from Skowron et al. 2023, Baliga et al. 2021, and Florin et al. 2019 in the discussion would enhance its impact. Additionally, providing evidence of Api binding to the involved structures would greatly bolster the work, as while the evidence supporting Api's involvement in ribosome synthesis impairment is convincing, the precise mechanism remains elusive and speculative.

> We are glad that reviewer #1 appreciates our work and finds the conclusions well-supported by evidence. In addition, we follow the suggestion and now discuss work from the Mankin group and others (lines 546-551).

Furthermore, in line with the reviewer's advice, we have made an effort and performed new experiments to provide direct evidence for Api137 binding to the precursor particles (new Fig. 7 and related text passages). The new cryo-EM structures demonstrate that Api137 has the capacity to interact with the respective precursor particles, supporting our hypothesis that Api137 can have a direct inhibitory effect on 50S assembly.

Lines 546-551: “Previous studies by the Mankin lab identified mutations in the 23S rRNA, ribosomal protein uL16, and RF1/RF2 of engineered B and K12 *E. coli* strains that rendered them less susceptible or partially resistant to Api137 and Api137 derived analogs, with the latter mostly related to mutations in RF^{13,69,70}. Skowron et al. reported one promising apidaecin derivative with multiple non-canonical amino acid substitutions able to circumvent the mentioned RF-dependent resistance⁷⁰, which leaves room for further studies in the assembly context.”

From a methodological standpoint, the paper appears solid and well-constructed. However, my limited expertise in cryoEM of the ribosome precludes me from offering a critical evaluation of the entire process.

1.2 The main (and maybe only significant) drawback of the paper, in my opinion, is indeed its highly descriptive nature, which demands a deep understanding of ribosome biology for full comprehension and critic analysis. Consequently, the paper as it is, may be only partially accessible to a broad audience. Authors may consider if the paper may benefit from submission to a more specialized journal focused on structural biology.

> We feel that Nature Communications is an ideal platform for our work, as it reaches a broad audience with a keen interest in structural studies. Nevertheless, we are grateful for reviewer #1’s advice and added more explanations (lines 123-127) and extended the discussion (line 404-408) to better communicate the importance of our findings to a general audience.

With this in mind, we have revised the entire manuscript, making changes and adding explanations throughout, which are highlighted in blue.

Lines 122-126: „We note that our strategy only detects precursors that have already incorporated the tagged proteins. Based on previous studies^{36,48}, both bL19 and bS20 assemble early during the biogenesis of the large and small subunit, respectively⁴⁸. Hence, this setup allowed monitoring and quantification of potential precursors of both the 50S and 30S ribosomal subunits. For precursors of the 30S subunit, however, we found no evidence. “

Lines 403-407: „Several structural studies focusing on the assembly of the bacterial large ribosomal subunit revealed the order of events and interdependencies, allowing a better understanding of the process from a mechanistic point of view^{21,35–37,39,42,47,53}. An important motivation for gaining insight into the process of ribosome assembly is the development of molecules inhibiting the ribosome assembly, which is considered an attractive antimicrobial drug target^{54,55}. “

In addition, our study now provides experimental evidence (see point 1.1) that Api137 has the capability to directly interfere with 50S assembly by preventing incorporation of ribosomal proteins and inducing misfolding of ribosomal components.

MINORS:

1.3 Fig 1D, the profile of MC4100 show diffused bandshift compared to RN31. Overall the protein

profiles are clear and the profiles can be compared easily, but maybe authors may mention this minor point.

> We thank reviewer #1 for pointing out this issue. To address this minor point, the gel was repeated with aliquots of 70S ribosomes isolated from MC410 and RN31. The two lanes now exhibit the same band pattern (except from the two fusion proteins in RN31) and can be compared easily (see Fig.1d).

1.4 line 167. "...with other PrAMPs" should be in my opinion tuned down, as the comparison has been done with Api88 and Onc112. As it is, the sentence sounds bit too generic with respect to what it has been done.

> We agree and toned down our claim accordingly.

Line 177: "...with the PrAMPs Api88 and Onc112...".

1.5 Line 231. Fix the missing reference

> We have fixed this erroneous Figure reference.

1.6 Line 263. Fix the missing reference

> We have fixed this erroneous Figure reference.

We thank reviewer #1 for the constructive criticism.

Reviewer #2 (Remarks to the Author):

The manuscript analyzes the presence, composition, and structures of accumulated assembly intermediates of the 50S ribosomal subunit when *E. coli* growth is inhibited by Api137, a translation-inhibiting antibacterial peptide derived from honey bees. Comparison with two controls (no treatment, treatment with Onc112 - another translation-inhibiting peptide derived from a milkweed bug) suggests that the 50S assembly intermediates observed are specific to Api137 treatment. Single-particle cryo-EM structure analysis and subsequent image classification is used to characterize the different structural states within the 50S assembly intermediate population. Specific changes such as absence/loss of the proteins uL22, uL23, uL29, and bL32 and structural changes in ribosomal RNA (misalignment of helix 61 (H61) are found, characterized, and interpreted.

The main finding of the work that Api137 treatment results in ribosome assembly problems seems well-supported by the data presented. I do have a few questions and concerns about the results and interpretations in this work:

2.1 All of the PDB validation reports seem to have '**not for manuscript review**' written on them. Are none of the structures deposited in the RCSB databank? This might mean that the oversight provided by the PDB in quality control of the structures is missing at this time.

> We appreciate the reviewer's attention to this detail. The final validation reports are only obtained upon release of the structure after publication. In the submission phase, we provided the models, maps, and validation reports for thorough examination. We have submitted the PDB depositions and have added the corresponding EMD and PDB accession codes (Lines 840 and following). In addition, we have generated final validation reports and provide them. Upon acceptance of the manuscript, we will promptly finalize the release process with the RCSB, ensuring full compliance with the required quality control standards.

Lines 840 and following:

Cryo-EM density maps and atomic models of pre50S precursors from Api137-treated cells are stored in EMDB and PDB as follows:

d126_(L29)-(L22)- (EMD-51828/ PDB ID 9H3K), *C_(L29)-(L22)-* (EMD-51829/ PDB ID 9H3L), *C_(L22)-* (EMD-51830/ PDB ID 9H3M), *C_(L22)-~H61* (EMD-51831/ PDB ID 9H3N), *C_GAC_(L22)-* (EMD-51832/ PDB ID 9H3O), *C-CP_(L22)-* (EMD-51833/ PDB ID 9H3P), *C_YjgA_(L22)-* (EMD-51834/ PDB ID 9H3Q), *C_YjgA_(L22)-~H61* (EMD-51835/ PDB ID 9H3R), *C_YjgA* (EMD-51836/ PDB ID 9H3S), *C_L2* (EMD-51837/ PDB ID 9H3T), *C_L2/H68* (EMD-51838/ PDB ID 9H3U), *C-CP_L2/L28* (EMD-51839/ PDB ID 9H3V), *C-CP_L2-H68* (EMD-51840/ PDB ID 9H3W), *C-CP_L2/L35-H68* (EMD-51841/ PDB ID 9H3X), *50S_(L16)-* (EMD-51842/ PDB ID 9H3Y), *50S* (EMD-51843/ PDB ID 9H3Z)

2.2 There seem to be 8 ribosome structures previously deposited in the RCSB databank with Api137 bound in the NPET (PDB IDs: 5O2R, 6GXN, 6GXO, 6GXP, 6GWT, 6YSS, 6YST, 6YSU).

In all these structures, the Api137 peptide is bound in virtually the same location. The authors refer to another study of their own and indicate that it is **accepted for publication**, where they have identified another binding site for Api137 (sometimes accompanied by a mention of reference 13, which is a Api137 structure publication from 2017). There are two panels in Supplementary figure 6 that seem to show a second binding site of Api137 but it is not possible to assess the validity/implications of the second site through data presented here.

> We are sorry for the confusion regarding Api137 binding sites. Our mentioned paper identifying additional binding sites has been published in the meantime (Nat Commun. 2024 May 10;15(1):3945), and we have updated the reference. The additional Api137 binding site is located at the exit site of the polypeptide exit tunnel. Our new data (new Fig. 7) indicate that the interaction of Api137 with the alternative binding site near the exit pore can already occur during assembly, as suggested by the Api137 density observed in the precursor states $C_{(L22)^-}$ and $C_{(L22)^-}\sim H61$. Consequently, this interaction may inhibit the progress of large subunit assembly. A corresponding section (Api137 binds to purified precursors) is included in the manuscript.

Lines 356-401:

Recently, we identified a second binding site for Api137 on the 50S, located at the PET exit pore in close proximity to the ribosomal proteins uL22 and uL29 (Supplementary Figure 5, Supplementary Figure 7)³⁴. Since none of the precursors, 50S or 70S ribosomes, revealed a defined density for Api137, either at its canonical binding site within the PET (Supplementary Figure 11a-c) or at its binding site at the PET exit pore (Supplementary Figure 12e), we considered the possibility that it had dissociated during sample preparation, possibly during the sucrose gradient purification step. To test this hypothesis and provide direct evidence of Api137's ability to bind to the precursors, they were incubated for 20 minutes with 30 $\mu\text{mol/L}$ Api137, representing a 100-fold excess relative to the ribosomal fraction. Precursors were plunge-frozen, and a high-resolution cryo-EM dataset was acquired and processed (Supplementary Figure 10).

Now, the 50S class showed a clear extra density for Api137 at the canonical PET binding site (Supplementary Figure 11e) and a defined extra density at the PET exit pore site, corresponding to a second Api137 molecule, as observed previously³⁴. However, even late precursors did not show Api137 density at the PET binding site within the tunnel (Supplementary Figure 11d)^{13,34}, most probably because the PTC matures last during 50S assembly^{35,37}. In contrast, the second Api137 binding site at the PET exit pore site was found occupied in the precursor states. After class identification, states were grouped into $dI26_{(L22)^-}/(L29)^-$, $C_{(L22)^-}$, $C_{(L22)^-}\sim H61$, C_{L22} , $C\text{-}CP_{(L22)^-}$, $C\text{-}CP_{(L22)^-}\sim H61$, and $C\text{-}CP_{L22}$ classes, 50S and 70S classes, and pooled again to increase particle numbers with a specific organization at the PET exit pore site (Figure 7). An additional focused classification step was performed on the pooled classes with a fragmented or absent density for uL22, using a local mask on the uL22, Api137, and uL29 binding sites to ensure that uL22 was absent in all particles of this subset. Atomic models of Api137 within the PET and at the PET exit site from our previous high resolution 50S structure³⁴ were rigid-body fitted into each of these pooled states.

Classes that showed both uL22 and uL29 densities exhibited a defined Api137 density at the exit pore binding site (Lauer et al., 2024) (Figure 7c, Supplementary Figure 12). In addition, both the $(L22)^{-} \sim H61$ class and $(L22)^{-}$ class showed a less defined density for Api137, indicating that the peptide can bind even in the absence of uL22 (Figure 7a, b) but remains more flexible. In the $(L22)^{-}$ states, Api137 showed alternative C-terminal trajectories (Figure 7d, Supplementary Figure 12e). Two trajectories could be followed, one of which showed that the peptide projects linearly toward the uL22 binding site (Figure 7e). Interestingly, this conformation would sterically clash with uL22 if it were present. No density was observed for classes that lacked both the early L-proteins uL22 and uL29, together with domain III.

Taken together these data indicate that Api137 can interact with pre-50S precursors as soon as the PET exit binding site is established. This interaction potentially blocks the incorporation of uL22 and provides a mechanistic explanation for direct inhibition of ribosome assembly by Api137.

2.3 The number of particles in some of the cryo-EM density reconstructed classes is quite small. It is not clear if an intermediate ab initio reconstruction step has been used in generating the final cryo-EM density using only the particle images for each class. If not, then it is necessary to do so. This will demonstrate that each class has sufficient particle image views to be reconstructed without a reference bias introduced by provision of a starting structure.

> We acknowledge the reviewer's concern regarding the small number of particles in some of the cryo-EM density reconstructed classes. We had initially performed heterogeneous refinements following 3D variability clustering to validate the classes, as shown in Supplementary Figure 3 (Sorting schemes, indicated by 3D variability clustering and heterogeneous refinement).

However, we appreciate the reviewer's valuable recommendation and proceeded with an ab initio reconstruction for each final class. Here, we noticed that one class ($d126_{(L29)^{-}}/(L22)^{-}$) needed to be subjected to another clean-up round using ab-initio reconstruction (2 classes), followed by heterogeneous refinement, as it still contained non-informative particles. We have added this specific sorting step in Supplementary Figure 3. Now, all identified states could be recapitulated using a single class ab-initio reconstruction.

We provide the results of the ab-initio reconstructions in the revised version of the manuscript. Additionally, we have combined this step with suggestion 2.4 of reviewer #2 and have created Supplementary Figure 6, where we have fitted the structural models to the maps generated using ab-initio reconstructions.

2.4 The cryo-EM densities reconstructed from the various particle classes have a range of resolutions. To rule out the possibility that certain protein densities are not visible only due to higher threshold values used for display, cartoon/ribbon models should be shown within their corresponding transparent cryo-EM densities, with each cryo-EM density displayed at a common threshold value with no hiding of the dust/noise (perhaps in a supplementary figure?).

> We appreciate the reviewer's suggestion. In the new Supplementary Figure 6, cartoon/ribbon models are displayed within their corresponding transparent cryo-EM densities. Each cryo-EM

density map is scaled to a standard deviation of 1, with no filtering or hiding of the dust/noise and set to a common threshold of 0.33.

In the first version of the manuscript, we referred to additional heterogeneity in some classes, which could not be further sorted due to low particles numbers.

Lines 781-784: “Classes were sorted until no more variability was observed or until particle counts were too low. Several sparsely populated classes still showed well resolved features which appeared sub-stoichiometric indicating residual heterogeneity.”

We have now included statements on specific heterogeneity observed for each subclass in the Figure legend of Supplementary Figure 6:

“Remaining heterogeneity in parts of the individual states is indicated in **bold**. **a** *d126_(L29)/(L22)*: (**uL6, bL30**), **b** *C_(L29)/(L22)*: (**uL6, bL30**), **c** *C_(L22)*: (**H34**), **d** *C_(L22)~H61*: (**bL30**), **e** *C_GAC(L22)*: (**bL30, H34, H63**), **f** *C-CP_YjgA_(L22)~H61*, **g** *C-CP_YjgA_(L22)*, **h** *C-CP_YjgA: YjgA*, **i** *C_L2*: (**uL6, uL22, bL30**), **j** *C_H68*: (**uL6, bL30, H90-93**), **k** *C-CP_L2/L28*, **l** *C-CP_(L22)*: (**uL22**), **m** *C-CP_H68*: (**H90-93**), **n** *C-CP_H68_L35*: (**H90-93**), **o** *50S_(L16)* **p** *50S* “

In an additional, new data set, we supplemented precursors derived from Api137-treated cells with Api137 (see new Figure 7, Supplementary Figure 10). Here, we collected a larger number of ribosomal particles (625.899 particles vs 446.030 particles). This resolved more of the heterogeneity mentioned above.

2.5 Were the three cultures (Api137 treated, Onc112 treated, and no treatment) grown simultaneously such that differences due to all other experimental variables would be minimized? Whether so or whether they were temporally distinct experiments should be clarified in the methods section.

> We clarified the procedure in the methods section (line 627): “The bacterial cultures treated with Onc112, Api88, erythromycin or chloramphenicol were always grown along with cultures treated with water and Api137.

2.6 The S20-mcherry and L19-EGFP fluorescence assessment of pre-50S populations are fine, but these can only assess populations that have these proteins present. This assessment will therefore miss any assembly intermediates without these proteins. This should be stated in the manuscript.

> We thank reviewer #2 for bringing up this point. We now explain the limitations of bS20 and bL19 labeling and provide arguments that speak in favor of them.

Lines 122-126: “We note that our strategy detects only those precursors that have already incorporated the tagged proteins. Based on previous studies^{36,48}, both bL19 and bS20 assemble early during the biogenesis of the large and small subunit, respectively. Hence, this setup allowed

monitoring and quantification of potential precursors of both the 50S and 30S ribosomal subunits. For precursors of the 30S subunit, however, we found no evidence.”

2.7 Is the EGFP density observed in all the 50S cryo-EM densities that have L19? If so, is the EGFP model fit into its corresponding density?

> This is an interesting point. However, EGFP density is not observed at all, because the EGFP molecule is attached to the surface exposed C-terminus of bL19 via a flexible linker, which leads to a high degree of mobility and precludes cryo-EM based detection of EGFP. Such flexible linkers are used to minimize adverse effects of the EGFP protein to folding and function of the fusion partner; in this case bL19. We added this information to the text.

Line 191-193: “In addition, the EGFP density is not observed because the EGFP molecule is attached to the surface-exposed C-terminus of bL19 via a flexible linker. This results in a high degree of mobility, which precludes its detection using cryo-EM.”.

2.8 On lines 180-181, where there is a disclaimer about absence of cryo-EM densities, there is the following line: 'However, at this stage of maturation, when it exhibits full density, its presence becomes unambiguous.' It is not clear which stage of maturation is being referred to and why the presence becomes unambiguous.

> This is a general warning describing the limitations of the method as such to deal with conformational heterogeneity and does not refer to a specific stage or situation. We have rephrased the passage and removed the last sentence, which may have caused the confusion:

Lines 187-191: “At this point, a word of caution is warranted. The absence of density in a cryo-EM map does not necessarily indicate the absence of the corresponding component, because it could be present in a potentially flexible or dynamic state. When we describe the binding or docking of a protein or an rRNA element to a precursor particle, we acknowledge the possibility that it may have been previously associated with the particle.”

2.9 On lines 194-195 (and Fig 3C), the proportion of 30S subunit particles remaining similar between untreated and Api137 treated samples seems to not agree with either the fluorescence data or the expectation that 30S subunits would accumulate in the absence of their combination with active 50S subunits to form 70S ribosomes. Is there an explanation for this discrepancy?

> Indeed, when focusing on the fluorescence signal, the 30S peak in the Api137 treated sample (h) is by far higher than in the untreated sample (f). Overall, the fractions 44-63 used for the cryo-EM analysis, which contain most of the pre-50S precursors, likely exhibit comparable amounts of co-migrating 30S subunits. While the early fractions (44-50) contain more 30S (indicated by red fluorescence) in the Api137-treated samples, the late fractions (55-63) show a higher abundance of 30S in the untreated samples. Figure 3c analyzes the particle distribution identified on the cryo-EM grids, providing a rough estimate. Some structural states may have a preferred orientation or excessive flexibility, which can prevent them from separating into 3D classes and being accurately counted. These considerations are now explained in the text.

Lines 206-211: "...remained relatively constant (Fig. 3c), which is in good agreement with fractions 44-63 of the ribosome profiles (Fig. 1f and h). We note that the number of particle images in cryo-EM classes is only a rough estimate of the relative populations in the sample. Some structural states may have a preferred orientation or excessive structural flexibility, preventing them from separating into 3D classes and being accurately counted. Nevertheless,..."

2.10 It should be made clear that the number of particle images in cryo-EM classes is only a rough estimate of relative populations in the sample. There could be structural states with a preferred orientation that do not separate into 3D classes and do not get correctly counted.

> We fully agree. This point is related to the previous (2.9). Either states in preferred orientation, or states exhibiting a too high degree of structural flexibility may escape our attention. We now included a corresponding statement, also see point 2.9.

Lines 208-211: "We note that the number of particle images in cryo-EM classes is only an imperfect estimate of the relative populations in the sample. Some structural states may have a preferred orientation or excessive structural flexibility, preventing them from separating into 3D classes and being accurately counted. Nevertheless,..."

2.11 The figures with structures are visually appealing but could be made clearer. In Fig 1, it is not immediately apparent whether the structures shown are structural models or cryo-EM densities. Also in Fig. 1, the 5S RNA is mentioned to be color coded as dark grey, but dark grey parts seem to appear in more than one region of the 50S subunit. It is my personal opinion that having a surface representation for one part of an atomic model with a ribbon representation for another makes it less clear to understand what is being shown, especially for a general readership, and using a mix of color and transparency with a consistent representation (especially ribbon) works better.

> We thank the reviewer for the thoughtful suggestions regarding the visualization of the figures. As a result of communications with the non-structural biologists involved in the project, we decided to use this specific visualization method to highlight structurally important variant regions clearly.

We agree that the representation could be made clearer and have therefore revised the legend of Fig. 3 to explicitly state "structural surface model" at 5 Å resolution. Also, we have separated statements on coloring and structural representation:

Lines 233-236: "States are shown as structural surface models at 5 Å resolution. Invariant parts between the pre-50S and 50S states are shown in light gray. Variant parts are color-coded according to the 23S rRNA domain architecture (domain II: cyan, domain IV: dark yellow, domain V: red, 5S rRNA: dark grey)."

While we have maintained the current visualization approach, we have made adjustments to ensure uniformity in representations: single variant helices are now consistently shown as colored ribbon models (including domain II helices), and domain III is displayed as a surface model to maintain visual clarity. We believe this way of presentation effectively balances detailed structural representation and readability for a broader audience.

2.12. I am not sure whether presence/absence of a specific proteins in 50S assembly intermediate cryo-EM densities or alternate orientation of helices can be considered as determinants of 'irreversible' or 'dead-end' nature for such intermediates. I think such definitive language might be better avoided.

> We agree and avoid the terms “irreversible” and “dead end”. The title now reads: The proline-rich antimicrobial peptide Api137 disrupts large ribosomal subunit assembly and induces misfolding

Line 34: “These data suggest a second mechanism for Api137,...”

Line 36 “...suggesting a bactericidal mechanism.”

line 462: While states in the “(L22)⁻~H6I” route...,”

In other cases (lines 282 and 488) we refer to **potential** dead-end precursors.

2.13 Is the Api137 density not seen in any of the fully-formed ribosome structures (e.g. 70S) in the Api137 treatment sample?

> Indeed, we did not observe Api137 density in either the 50S or 70S ribosomal subunits in the original preparation. To address the possibility that Api137 molecules may have dissociated during sample preparation (such as during lysis, sucrose gradient centrifugation, etc.), we conducted a new experiment where Api137 was supplemented to the precursor-containing preparation after gradient centrifugation (see new Figure 7). Following incubation, density corresponding to Api137 was observed at both at the exit pore (Fig. 7) and PTC binding sites (Supplementary Figs 11 and 12). In addition, see point 2.2.

2.14 The authors seem to suggest a direct steric mechanism for Api137 interference with ribosome assembly (line 433). Both the previously known Api137 binding sites and the novel ones mentioned by the authors are in properly assembled ribosomes. If the authors are proposing other unknown or non-specific binding modes of Api137 in unassembled ribosomal RNA, this should be clearly stated.

> We thank the reviewer for directing our attention to this issue. We now clearly state that we suggest a specific and direct mechanism for Api137 interference with ribosome assembly, based on the data derived from the new experiment mentioned in point 2.13 (new Fig. 7). Line 519: “...suggesting a specific and direct effect on early and late stages of large subunit assembly. ...”

Nevertheless, we point out in the discussion that the antibacterial effect of Api137 is probably based on interference with both, regular translation and direct inhibition of assembly.

Lines 541-544: “Since Api137 targets the highly conserved bacterial ribosome at multiple stages by interfering with both, its assembly and with termination of translation, the development of resistance due to mutations in ribosomal proteins and rRNA is supposed to be a low probability scenario. However, the extent to which each mechanism contributes to the observed antibacterial activity remains to be determined...”

2.15 Could the Fig. 2 title be altered to not mention 'ribosome profile' which has a different default interpretation these days. Perhaps something like 'Statistical ribosome sucrose gradient profile analysis ...' instead?

> We thank the reviewer for bringing up this point. In the Fig. 2 caption we now refer to “Statistical analysis of absorbance (A_{254}) and fluorescence-based ribosome profiles of *E. coli* RN31...”

2.16 Lines 110 and 129 refers to a URL about mass spectrometric data (https://panoramaweb.org/Api137_immature_ribo.url) that seems to be password-protected.

> We apologize for the malfunctional link. Reviewers can access the data using login details as follows: https://panoramaweb.org/Api137_immature_ribo.url
Email: panorama+reviewer251@proteinms.net
Password: 3t+vg6PpmWicNF.

2.17 In the pdf file I read, lines 231-232 and 263-264 have a '(Error! 231 Reference source not found.' message.

> We thank reviewer #2 for the hint and corrected the references.

We thank reviewer #2 for the constructive criticism.